# GoodDrag: Towards Good Practices for Drag Editing with Diffusion Models

**Zewei Zhang**[†]   **Huan Liu**[†]   **Jun Chen**[†]   **Xiangyu Xu**[*‡]
[†]McMaster University       [‡]Xi'an Jiaotong University

## Abstract

In this paper, we introduce GoodDrag, a novel approach to improve the stability and image quality of drag editing. Unlike existing methods that struggle with accumulated perturbations, GoodDrag introduces an AlDD framework that alternates between drag and denoising operations within the diffusion process, effectively improving the fidelity of the result. We also propose an information-preserving motion supervision operation that maintains the original features of the starting point for precise manipulation and artifact reduction. In addition, we contribute to the benchmarking of drag editing by introducing a new dataset, Drag100, and developing dedicated quality assessment metrics, Dragging Accuracy Index and Gemini Score, utilizing Large Multimodal Models. Extensive experiments demonstrate that the proposed GoodDrag compares favorably against the state-of-the-art approaches both qualitatively and quantitatively. The source code and data are available at https://gooddrag.github.io.

## 1 Introduction

In this work, we present GoodDrag, a novel approach for drag editing with enhanced stability and image quality. Drag editing (Pan et al., 2023) represents a new direction in generative image manipulation. It allows users to effortlessly edit images by simply specifying the starting and target points, as if physically dragging an object or a part of an object from its initial location to the target location, with the edits blending harmoniously into the original image context as exemplified in Figure 2.

Early methods (Pan et al., 2023; Ling et al., 2023) for drag editing employ Generative Adversarial Networks (GANs) (Goodfellow et al., 2014) which are often trained for class-specific images, and thereby struggle with generic, real-world images. Moreover, these methods heavily rely on GAN inversion techniques (Roich et al., 2022; Weihao et al., 2021; Xu et al., 2023), which may fail in complex, in-the-wild scenarios.

To address these issues, recent advancements have shifted towards using diffusion models for drag editing (Shi et al., 2023; Mou et al., 2024a; Nie et al., 2023; Mou et al., 2024a;b). Thanks to the remarkable capabilities of diffusion models in image generation, these methods have significantly improved the quality of drag editing for generic images. However, the current diffusion-based approaches often suffer from instability, resulting in outputs that have severe distortions or fail to adhere to the designated control points.

This paper addresses these challenges by establishing two good practices for more effective drag editing using diffusion models. Our first contribution is a new editing framework, called Alternating Drag and Denoising (AlDD). As shown in Figure 1, existing methods typically conduct all drag operations at once and then attempt to correct the accumulated perturbations subsequently. However, this approach often leads to perturbations that are too substantial to be corrected. In contrast, the AlDD framework alternates between the drag and denoising operations within the diffusion process as shown in Figure 1. This methodology effectively addresses the issue by preventing the accumulation of large distortions, ensuring a more refined and manageable editing process.

---

[*]Corresponding author: xuxiangyu2014@gmail.com

As the second contribution, we investigate into the common failures of point control, where the starting point cannot be accurately dragged to the desired target location. We find this is mainly due to that the dragged features in existing algorithms may gradually deviate from the original features of the starting point. To address this issue, we propose an information-preserving motion supervision operation that maintains the original features of the starting point, ensuring more realistic and precise point control.

Furthermore, we make early efforts to benchmark drag editing by introducing a new dataset Drag100 along with dedicated evaluation metrics. Notably, we develop Gemini Score, a novel quality assessment metric utilizing Large Multimodal Models (Anil et al., 2023), which is more reliable and effective than existing image quality assessment metrics.

Combining these good practices, our final algorithm, named GoodDrag, consistently achieves high-quality drag editing results and outperforms state-of-the-art approaches both quantitatively and qualitatively.

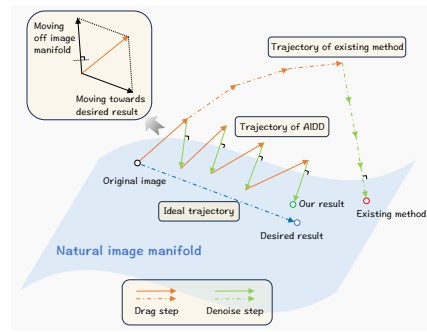

Figure 1: There are two main operations involved in drag editing: drag and denoising. The drag operation (orange) modifies the image to achieve the desired drag effect but leads to deviations from the natural image manifold, resulting in artifacts. The denoising operation (green) estimates the score function of the natural image distribution and corrects the artifacts by guiding the results back to the manifold. Existing diffusion-based drag editing methods (dotted trajectory) apply all drag operations at once, which often results in excessive and accumulated perturbations that are hard to correct. In contrast, the proposed AlDD framework (solid trajectory) alternates between drag and denoising within the diffusion process, which prevents accumulated perturbations and ensures more accurate results.

## 2 RELATED WORK

**Diffusion-based image manipulation.** Generative Adversarial Networks (GANs) have long dominated image editing tasks such as inpainting, colorization, super-resolution, and text-driven manipulation, demonstrating early success in synthesizing plausible visual content (Xu et al., 2017; Yu et al., 2018; Isola et al., 2017; Park et al., 2019; Chen et al., 2020; Xu et al., 2021; Chen et al., 2021; Chan et al., 2022; Liu et al., 2023b; Du et al., 2023). However, they often suffer from instability and suboptimal image quality for real-world input, hindering their application to complex, practical editing scenarios.

Recent breakthroughs in diffusion models have revolutionized the field, offering unparalleled image fidelity and controllability (Sohl-Dickstein et al., 2015; Ho et al., 2020; Song et al., 2020a;b; Rombach et al., 2022; Su et al., 2022; Liu et al., 2023a; Li et al., 2024; Yan et al., 2024; Dhariwal & Nichol, 2021). For instance, the Dreambooth series (Ruiz et al., 2023a; Raj et al., 2023; Ruiz et al., 2023b) leverage subject-specific tuning to edit and create new contents under the theme of the input. CustomSketching (Xiao & Fu, 2024) and ControlNet (Zhang et al., 2023) introduce sketches, text, and user scribbles to guide the generation of images. Instant3D (Li et al., 2024) further integrates geometric priors to push the boundary to 3D-consistent generation. These advancements underscore the potential of diffusion models in a wider range of image editing tasks, such as drag editing.

**Drag Editing** Drag editing, introduced by DragGAN (Pan et al., 2023), revolutionized image manipulation through intuitive point-to-point selection. Nevertheless, GAN-based drag editing approaches (Pan et al., 2023; Roich et al., 2022; Ling et al., 2023) remain fundamentally constrained: they primarily edit synthetic images and struggle with real-world scenes.

Recent diffusion-based methods (Shi et al., 2023; Nie et al., 2023; Mou et al., 2024a; Liu et al., 2024; Hou et al., 2024; Shin et al., 2024; Mou et al., 2024b; Lu et al., 2024; Zhao

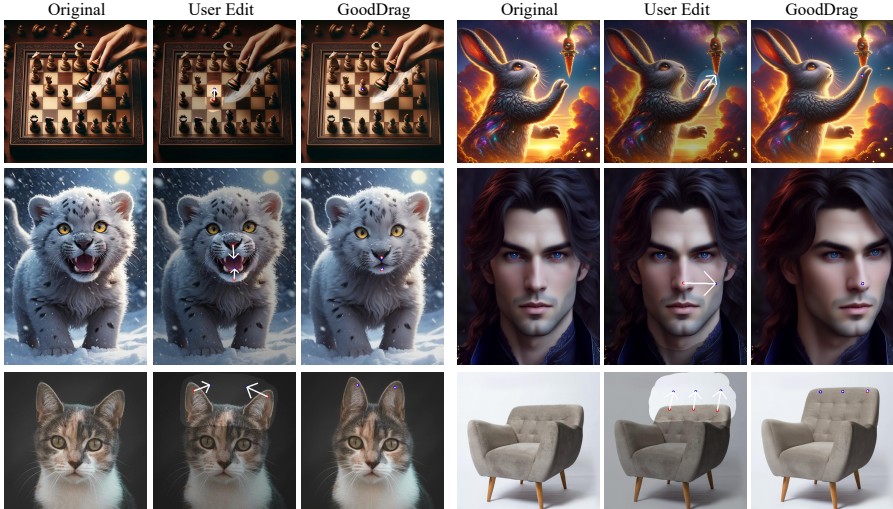

Figure 2: Given an input image (Original) and user-specified control points (User Edit), GoodDrag "drags" the semantic content from the handle points (red) to the target points (blue). The target points remain fixed while the handle points move closer during optimization. Users can also select an indication mask to define the editable region.

et al., 2024) enable drag editing on diverse real images. However, their consecutive editing steps accumulate perturbations in the latent space (Figure 1), which often lead to severe image quality degradation.

Our key contribution addresses this limitation through the Alternating-Drag-and-Denoising (AlDD) framework. Unlike existing approaches, AlDD strategically interleaves drag operations with denoising steps across the entire generation process, enabling progressive refinement of edits while preventing artifact accumulation. We further introduce an information-preserving motion supervision mechanism that mitigates feature drift, stabilizing the editing trajectory. These good practices ensure high-fidelity, artifact-free results even for complex real-world images, advancing drag editing beyond the quality and generality constraints of existing methods.

## 3 METHOD

In this work, we propose GoodDrag, a new framework, for high-quality drag editing with diffusion models (Song et al., 2020a;b; Rombach et al., 2022). We develop and integrate two effective practices within this framework: Alternating Drag and Denoising (Section 3.2) and Information-Preserving Motion Supervision (Section 3.3), which are instrumental in reducing visual artifacts and enhancing precision in drag editing.

### 3.1 PRELIMINARY ON DIFFUSION MODELS

Diffusion models represent a compelling subclass of generative models, having demonstrated remarkable performance in synthesizing high-quality images, as evidenced by advanced applications like DALLE2 (Ramesh et al., 2022) and Stable Diffusion (Rombach et al., 2022). These models consist of two distinct phases: the forward process and the reverse process.

In the forward process, a given data sample $z_0$ is combined with increasing levels of Gaussian noise over a series of $T_{\max}$ steps. This process results in the generation of a series of progressively noised samples $\{z_t\}_{t=1}^{T_{\max}}$, with each $z_t$ representing the noised image at time step $t$. Mathematically, the forward process can be formulated as:

$$z_t = \sqrt{\alpha_t} z_0 + \sqrt{1 - \alpha_t} \varepsilon, \tag{1}$$

where $\varepsilon \sim \mathcal{N}(0, \mathbf{I})$ is a random Gaussian noise. $\alpha_t \in (0, 1)$ acts as a diminishing factor of $z_0$, and the sequence $\{\alpha_t\}_{t=1}^{T_{\max}}$ is designed to be monotonically decreasing for a stronger noise

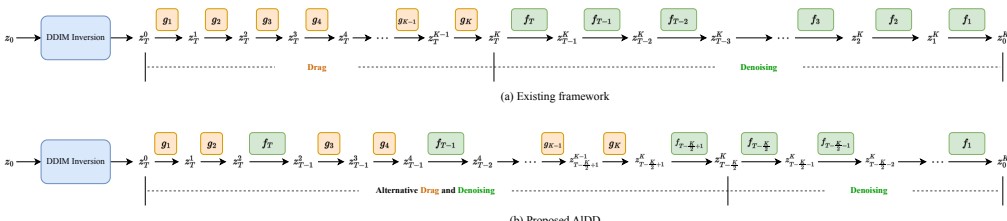

Figure 3: Overview of the proposed AlDD framework. (a) Existing methods first perform all drag editing operations $\{g_k\}_{k=1}^K$ at a single time step $T$ and subsequently apply all denoising operations $\{f_t\}_{t=T}^1$ to transform the edited image $z_T^K$ into the VAE image space. (b) To mitigate the accumulated perturbations in (a), AlDD alternates between the drag operation $g$ and the diffusion denoising operation $f$, which leads to higher quality results. Specifically, we apply one denoising operation after every $B$ drag steps and ensure the total number of drag steps $K$ is divisible by $B$. Here $T = T_{\max} \cdot \kappa$, where $\kappa$ is the DDIM inversion strength. We set $B = 2$ in this figure for a clear illustration and use $B = 10$ in real implementation.

as $t$ increases. When $t$ is close to $T_{\max}$, $\alpha_t$ is close to 0, and $z_t$ approximates an isotropic Gaussian distribution.

During the reverse process, we first sample $z_{T_{\max}}$ from the standard Gaussian distribution $\mathcal{N}(0, \mathbf{I})$ and then generate samples resembling the original data distribution of $z_0$ by gradually reducing the noise levels. The Denoising Diffusion Implicit Models (DDIM) (Song et al., 2020a) stand out in this phase, achieving decent efficiency and consistency in generating high-quality images. The reverse process from $z_t$ to $z_{t-1}$ under the deterministic DDIM framework can be written as:

$$z_{t-1} = \sqrt{\alpha_{t-1}} \frac{z_t - \sqrt{1 - \alpha_t}\varepsilon_\theta(z_t, t)}{\sqrt{\alpha_t}} + \sqrt{1 - \alpha_{t-1}}\varepsilon_\theta(z_t, t), \qquad (2)$$

where $\varepsilon_\theta$ represents a neural network with parameters $\theta$, which is trained to predict the noise $\varepsilon$ in Eq. 1. For clarity, we denote Eq. 2 as $z_{t-1} = f_t(z_t)$.

Following Stable Diffusion (Rombach et al., 2022), we use the Variational Autoencoder (VAE) (Esser et al., 2021) to encode original images into lower-resolution images in feature space to reduce computation and memory costs. Throughout the paper, the variables denoted by $z$ refer to images in this VAE space instead of the pixel space.

## 3.2 ALTERNATING DRAG AND DENOISING

> *"A stitch in time saves nine."*
>
> — Proverb

The input of drag editing is a source image $z_0$, a set of $l$ starting points $\{\boldsymbol{p}_i\}$, and their corresponding target points $\{\boldsymbol{q}_i\}$, where $i = 1, 2, \cdots, l$. Here, $\boldsymbol{p}_i, \boldsymbol{q}_i \in \mathbb{R}^2$ represent 2D pixel coordinates within the image plane. An optional binary mask M can also be provided to specify the image region that is allowed for edits. The objective of drag editing is to seamlessly transfer content from each starting point $\boldsymbol{p}_i$ to the designated target point $\boldsymbol{q}_i$, while ensuring that the resulting image remains natural and cohesive, with the edits blending harmoniously into the original image context.

The drag editing starts by transforming the source image $z_0$ into a latent representation $z_T$ through the DDIM inversion (see Appendix B), where the timestep $T$ is empirically chosen, typically close to $T_{\max}$. With the transformed $z_T$, the input image can be edited through a $K$-step iterative process as shown in Figure 3. Each iteration, denoted by $g_k$, $k = 1, \cdots, K$, comprises two main phases: motion supervision and point tracking (Pan et al., 2023; Shi et al., 2023; Ling et al., 2023).

Existing methods suffer from low image fidelity because they perform all drag operations within a single diffusion time step as shown in Figure 3(a), leading to accumulated perturba-

tions and distortions. To address this issue, we propose an Alternating Drag and Denoising (AlDD) framework. The AlDD distributes editing operations across multiple diffusion time steps by alternating between drag and denoising steps, allowing for more manageable and incremental changes. As illustrated in Figure 3(b), after applying $B$ drag operations $g$ at time step $t$, a denoising step $f$ follows, converting the latent representation from $t$ to $t-1$ and alleviating artifacts from the drag step. This pattern continues at each subsequent time step until all intended drag edits are completed.

**AlDD motion supervision.** We denote the output of the $k$-th iteration, which serves as the input for the $(k+1)$-th iteration, as $z_t^k$ and the corresponding handle points as $\boldsymbol{p}_i^k$, with the initial image $z_T^0 = z_T$ and the initial handle point $\boldsymbol{p}_i^0 = \boldsymbol{p}_i$. The aim of motion supervision is to progressively edit the current image $z_t^k$ to move the handle points $\boldsymbol{p}_i^k$ towards their targets $\boldsymbol{q}_i$. Specifically, denoting the movement direction for the $i$-th point as $\boldsymbol{d}_i^k = \frac{\boldsymbol{q}_i - \boldsymbol{p}_i^k}{\|\boldsymbol{q}_i - \boldsymbol{p}_i^k\|_2}$, the motion supervision is realized by aligning the feature of $z_t^k$ around point $\boldsymbol{p}_i^k + \beta \boldsymbol{d}_i^k$ to the feature around $\boldsymbol{p}_i^k$, where $\beta$ is the step size of the movement. The feature of $z_t^k$ can be written as $\mathrm{F}(z_t^k) = \mathcal{I}\left(\mathrm{U}_\theta(z_t^k; t)\right)$, where the feature extractor $\mathrm{U}_\theta$ is the U-Net of Stable Diffusion parameterized by $\theta$, and $\mathcal{I}$ represents the interpolation function to adjust the feature map to the size of the input image.

The feature alignment loss for motion supervision in AlDD is defined as:

$$\mathcal{L}(z_t^k; \{\boldsymbol{p}_i^k\}) = \sum_{i=1}^{l} \left\| \mathrm{F}_{\Omega(\boldsymbol{p}_i^k + \beta \boldsymbol{d}_i^k, r_1)}(z_t^k) - \mathrm{sg}\left(\mathrm{F}_{\Omega(\boldsymbol{p}_i^k, r_1)}(z_t^k)\right) \right\|_1 \tag{3}$$
$$+ \lambda \left\| \left(z_{t-1}^k - \mathrm{sg}\left(z_{t-1}^0\right)\right) \odot (1 - \mathrm{M}) \right\|_1.$$

where $\Omega(\boldsymbol{p}_i^k, r_1) = \{\boldsymbol{p} \in \mathbb{Z}^2 : \|\boldsymbol{p} - \boldsymbol{p}_i^k\|_\infty \leqslant r_1\}$ describes a square region centered at $\boldsymbol{p}_i^k$ with a radius $r_1$. $\mathrm{sg}(\cdot)$ denotes the stop-gradient operation. The first term of Eq. 3 essentially drives the appearance of the image around $\boldsymbol{p}_i^k + \beta \boldsymbol{d}_i^k$ to get closer to the appearance around $\boldsymbol{p}_i^k$. The second term ensures the non-editable region, as indicated by $1 - \mathrm{M}$, remains unchanged throughout the editing process. Since the image $z_t^k$ has undergone $\lfloor \frac{k}{B} \rfloor$ denoising operations, we apply the drag operation at the diffusion time step $t = T - \lfloor \frac{k}{B} \rfloor$. This is in sharp contrast to existing methods, which apply all drag operations at a single time step $T$.

The motion supervision for the $(k+1)$-th iteration takes one gradient descent step according to the feature alignment loss $\mathcal{L}(z_t^k; \{\boldsymbol{p}_i^k\})$:

$$z_t^{k+1} = z_t^k - \eta \cdot \frac{\partial \mathcal{L}(z_t^k; \{\boldsymbol{p}_i^k\})}{\partial z_t^k}, \tag{4}$$

where $\eta$ is the step size. For short, we write the $(k+1)$-th drag step Eq. 4 as $z_t^{k+1} = g_{k+1}(z_t^k)$.

**Point tracking.** While the motion supervision effectively guides the movement of the handle point towards $\boldsymbol{p}_i^k + \beta \boldsymbol{d}_i^k$, its final position at this exact spot is not guaranteed. This necessitates the point tracking to locate the new location of the handle point $\boldsymbol{p}_i^{k+1}$, which is formulated as:

$$\boldsymbol{p}_i^{k+1} = \operatorname*{argmin}_{\boldsymbol{p} \in \Omega(\boldsymbol{p}_i^k, r_2)} \left\| \mathrm{F}_{\boldsymbol{p}}(z_t^{k+1}) - \mathrm{F}_{\boldsymbol{p}_i^0}(z_t^0) \right\|_1. \tag{5}$$

Eq. 5 identifies the updated handle point by searching the location in $z_t^{k+1}$ that most closely resembles the original starting point $\boldsymbol{p}_i^0$ in the original image $z_t^0$ based on feature similarity. $r_2$ denotes the radius of the search area $\Omega(\boldsymbol{p}_i^k, r_2)$.

Finally, we conduct the remaining denoising steps to convert the latent representation to the desired VAE image space $z_0$. Notably, the AlDD only changes the order of the computations, which improves editing quality without introducing additional computational overhead.

The key insight behind this framework is that addressing perturbations incrementally as they arise, rather than allowing them to accumulate, facilitates more effective and manageable image editing. In other words, it is better to fix the problem when it is small than to wait until it becomes more significant.

To validate this concept of AlDD, we conduct a toy experiment as shown in Figure 4. We simulate the perturbations introduced during image editing with random Gaussian noise, and compare the results of adding multiple noise samples within the same diffusion time step versus across different time steps. When noise is added all at once to $z_T$, the resulting image suffers from low fidelity as shown in Figure 4(b). This is due to the accumulation of noise within a single time step, leading to a substantial deviation from the image manifold. In contrast, distributing the noise across multiple diffusion steps results in well-corrected perturbations and better preservation of original content, as shown in Figure 4(c). This validates our hypothesis that progressive adjustments lead to more effective image editing.

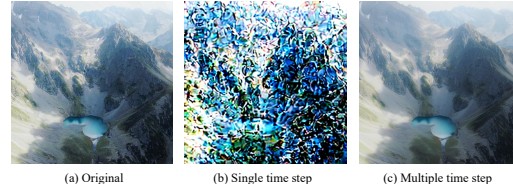

(a) Original     (b) Single time step     (c) Multiple time step

Figure 4: We generate 10 random noise samples from $\mathcal{N}(0, 0.1^2 \mathbf{I})$ and compare two scenarios: (b) We add all 10 noise samples to a single time step $z_T$ at once followed by 10 denoising steps, where the resulting image exhibits significant degradation. (c) We distribute the 10 noise samples across 10 different time steps, from $z_T$ to $z_{T-9}$, with a denoising step following each noise to prevent the accumulation effect; the resulting image better preserves the original content with higher fidelity.

### 3.3 INFORMATION-PRESERVING MOTION SUPERVISION

Another challenge in existing drag editing methods is the feature drifting of handle points, which can lead to artifacts in the edited results and failures in accurately moving handle points as shown in Figure 5(b). The feature drifting issue is illustrated in Figure 5(d)-(e), where the initial handle points (red points) in Figure 5(d) are near the boundary of the beach wave. As the number of drag steps increases, the handle points become less similar to their original appearance, drifting away from the wave boundary towards the sea foam or the sand, as shown in Figure 5(e).

We identify that the root cause of handle point drifting lies in the design of the motion supervision loss, as methods in (Pan et al., 2023; Shi et al., 2023; Ling et al., 2023). Their loss function encourages the next handle point, $\boldsymbol{p}_i^k + \beta \boldsymbol{d}_i^k$, to be similar to the current handle point, $\boldsymbol{p}_i^k$. Consequently, even minor drifts in one iteration can accumulate over time during motion supervision, leading to significant deviations and distorted outcomes.

To address this problem, we propose an information-preserving motion supervision approach, which maintains the consistency of the handle point with the original point throughout the editing process. The updated feature alignment loss for motion supervision is for-

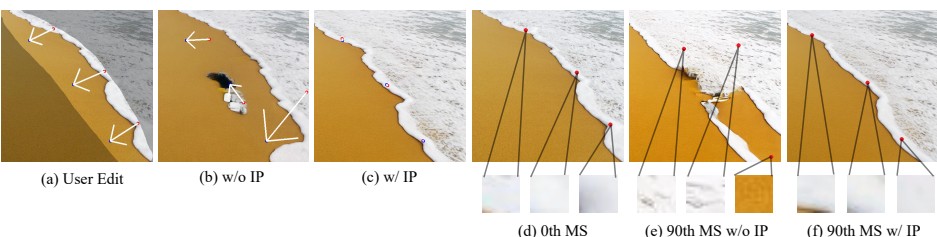

(a) User Edit     (b) w/o IP     (c) w/ IP     (d) 0th MS     (e) 90th MS w/o IP     (f) 90th MS w/ IP

Figure 5: Illustration of the feature drifting issue. In (d), the initial handle points are near the beach wave boundary. As drag editing progresses, the features of the handle points deviate from their original appearance. By the 90th motion supervision (MS) step shown in (e), the handle points have drifted away from the wave boundary, leading to artifacts and inaccurate movement in (b). To address this issue, we propose information-preserving motion supervision (IP) to maintain the fidelity of handle points to their original appearance (f), resulting in higher-quality results (c).

Figure 6: Distribution of categories and tasks in the Drag100, along with example images and user edits.

mulated as:

$$\mathcal{L}(z_t^k; \{\boldsymbol{p}_i^k\}) = \sum_{i=1}^{l} \left\| F_{\Omega(\boldsymbol{p}_i^k + \beta \boldsymbol{d}_i^k, r_1)}(z_t^k) - \text{sg}\left( F_{\Omega(\boldsymbol{p}_i^0, r_1)}(z_t^0) \right) \right\|_1$$
$$+ \lambda \left\| \left( z_{t-1}^k - \text{sg}\left( z_{t-1}^0 \right) \right) \odot (1 - \text{M}) \right\|_1 , \tag{6}$$

where $\boldsymbol{p}_i^0$ is the original handle point in the unedited image $z_t^0$. This formulation ensures that the intended handle point $\boldsymbol{p}_i^k + \beta \boldsymbol{d}_i^k$ in the edited image $z_t^k$ remains faithful to the original handle point, thereby preserving the integrity of the editing process.

While the information-preserving motion supervision effectively addresses the handle point drifting issue, it introduces new challenges. Specifically, Eq. 6 is more difficult to optimize due to its inherently larger feature distance than the original motion supervision loss Eq. 3. Therefore, a straightforward application of Eq. 6 often results in unsuccessful drag effects of the handle point. Initially, we attempted to overcome this by increasing the step size $\eta$ in the motion supervision process (Eq. 4), which turned out to be unsuccessful. Instead, we find that maintaining a small step size and increasing the number of motion supervision steps before each point tracking offers a better solution:

$$z_{t,j+1}^k = z_{t,j}^k - \eta \cdot \frac{\partial \mathcal{L}(z_{t,j}^k; \{\boldsymbol{p}_i^k\})}{\partial z_{t,j}^k}, \quad j = 0, \cdots, J - 1, \tag{7}$$

where $z_{t,0}^k = z_t^k$ is the initial image, and $z_t^{k+1} = z_{t,J}^k$ is the output after $J$ gradient steps.

The proposed information-preserving motion supervision marks an effective practice for drag editing, which ensures that the handle point remains close to its original appearance without introducing excessive artifacts as shown in Figure 5(f). Consequently, this leads to higher-quality results, as evidenced in Figure 5(c).

Finally, the whole pipeline of GoodDrag is summarized in Algorithm 1 in the Appendix.

## 4 BENCHMARK

To benchmark the progress in drag-based image editing, we introduce a new evaluation dataset named Drag100, and two dedicated quality assessment metrics, DAI and GScore.

### 4.1 DRAG100 DATASET

Since drag-based image editing is still a nascent research area, there is a lack of evaluation datasets. While recent works have introduced two datasets (Shi et al., 2023; Nie et al., 2023), they have certain limitations. First, Nie et al. (2023) provides masks M for only a few samples, which can lead to uncontrolled experiments and difficulties in benchmarking and fair comparison of different methods. Second, these datasets were not constructed with explicit consideration for diversity in drag tasks, making evaluations less comprehensive.

To overcome these challenges, we introduce a new dataset called Drag100. This dataset consists of 100 images, each with carefully labeled masks and control points, ensuring that different methods can be evaluated in a controlled manner. Drag100 is designed to encompass a diverse range of content, as shown in Figure 6. It comprises 85 real images and 15 AI-generated images using Stable Diffusion. The dataset spans various categories, including 58 animal images, 5 artistic paintings, 16 landscapes, 5 plant images, 6 human portraits, and 10 images of common objects such as cars and furniture.

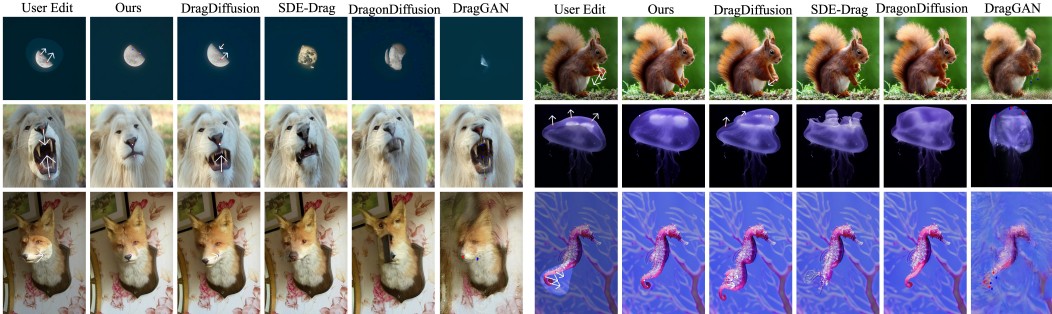

Figure 7: Comparison with drag editing methods (Shi et al., 2023; Nie et al., 2023; Mou et al., 2024a; Pan et al., 2023).

We have particularly considered the diversity of drag tasks, including relocation, rotation, rescaling, content removal, and content creation, as illustrated in Figure 6. These tasks have distinct characteristics. Relocation involves moving an object or a part of an object, while rotation adjusts the orientation of objects; both tasks mimic rigid motion in the physical world without changing the object area or creating new contents. Rescaling corresponds to enlarging or shrinking an object. Content removal involves deletion of specific image components, *e.g.*, closing mouth, whereas content creation involves generating new content not present in the original image, *e.g.*, opening mouth. These tasks require advanced hallucination capabilities, similar to occlusion removal (Liu et al., 2020) and image inpainting (Yu et al., 2018). By encompassing these diverse tasks, the Drag100 dataset enables a more comprehensive evaluation of drag editing algorithms.

## 4.2 Evaluation Metrics for Drag Editing

In this work, we introduce the following two quality assessment metrics, Dragging Accuracy Index (DAI) and Gemini Score (GScore), for quantitative evaluation.

**DAI.** Existing methods use Mean Distance (MD) (Shi et al., 2023) to measure the drag accuracy. While effective, it necessitates handle point identification with DIFT (Tang et al., 2023), which significantly increases runtime and imposes high demands on GPU resources. Specifically, MD requires 1.8s per image on Drag100 when using an A100 GPU. This high computational overhead makes it less practical for evaluating drag editing algorithms.

To address this issue, we introduce DAI to quantify the effectiveness of an approach in transferring the semantic contents to the target point. Specifically, we define DAI to assess whether the source content at $\boldsymbol{p}_i$ of the original image has been successfully dragged to the target location $\boldsymbol{q}_i$ in the edited image, which can be written as:

$$\text{DAI} = \frac{1}{l} \sum_{i=1}^{l} \frac{\left\| \phi(z_0)_{\Omega(\boldsymbol{p}_i, \gamma)} - \phi(\hat{z}_0)_{\Omega(\boldsymbol{q}_i, \gamma)} \right\|_2^2}{(1 + 2\gamma)^2}, \tag{8}$$

where $\phi$ is the VAE decoder converting $z_0$ to the RGB image space, and $\Omega(\boldsymbol{p}_i, \gamma)$ denotes a patch centered at $\boldsymbol{p}_i$ with radius $\gamma$. Eq. 8 calculates the mean squared error between the patch at $\boldsymbol{p}_i$ of $\phi(z_0)$ and the patch at $\boldsymbol{q}_i$ of $\phi(\hat{z}_0)$. By varying the radius $\gamma$, we can flexibly control the extent of context incorporated in the assessment: a small $\gamma$ ensures precise measurement of the difference at the control points, while a large $\gamma$ encompasses a broader context; this serves as a lens to examine different aspects of the editing quality.

In contrast to MD, the proposed DAI metric is much more efficient, requiring only 0.01s per image on the Drag100 dataset. Additionally, DAI runs entirely on the CPU, eliminating the need for GPU resources. The efficiency of DAI makes it particularly valuable for drag editing research, where a fast and accessible metric is essential for iterative development and large-scale benchmarking.

**GScore.** Existing methods use Image Fidelity (IF) (Shi et al., 2023) to evaluate the perceptual quality of the edited images. However, we find that the IF metric is fundamentally

flawed as an evaluation measure for drag editing. IF is defined as 1-LPIPS between the drag-edited image and the input image, meaning it penalizes any changes to the image, even when such changes are necessary to achieve the desired editing. As a result, the metric rewards outputs that are identical or nearly identical to the input image, which contradicts the very purpose of drag editing. This inherent limitation is clearly demonstrated in Figure 10 of the Appendix, where GoodDrag achieves the best visual quality yet receives the worst IF score, underscoring the inability of IF to accurately evaluate the quality of meaningful drag edits.

While No-Reference Image Quality Assessment (NR-IQA) methods (Ke et al., 2021; Golestaneh et al., 2022; Chen et al., 2023) offer a way to assess image quality without reference images, they often rely on handcrafted features or are trained on limited image samples, which do not always align well with human perception.

To address this challenge, we introduce GScore, a new metric leveraging Large Multimodal Models (LMMs) trained on internet-scale vision-language data. GScore uses LMMs as evaluators by providing the edited and original images as references and prompt them to rate perceptual quality on a scale from 0 to 10, with higher scores indicating better quality. As shown in Figure 10, compared to IF, the proposed GScore metric better correlates with human perception, making it a more reliable and appropriate metric for evaluating drag editing algorithms.

In our experiments, we explored the use of both GPT-4V (Achiam et al., 2023) and Gemini (Anil et al., 2023) as evaluation agents. We find that the output from Gemini is more reliable and closely aligned with human visual judgment. Therefore, we select Gemini as the primary evaluation agent for assessing the quality of edited images in our work.

## 5  EXPERIMENTS

### 5.1  IMPLEMENTATION DETAILS

In our experiments, we use Stable Diffusion 1.5 (Rombach et al., 2022) as the base model and finetune its U-Net with LoRA (rank=16) to enhance image fidelity. We employ the Adam optimizer (Kingma & Ba, 2014) with a 0.02 learning rate. For the diffusion process, we set $T_{\max} = 50$ denoising steps, an inversion strength of $\kappa = 0.75$ (resulting in $T = T_{\max} \cdot \kappa = 38$), and no text prompt. Features for Eq. 6 are extracted from the last U-Net layer. In the AlDD framework, we set the motion supervision and point tracking radii to $r_1 = 4$ and $r_2 = 12$, respectively, with a drag size $\beta = 4$ and a mask loss weight $\lambda = 0.2$. We perform a total of $K = 70$ drag operations, with $B = 10$ operations per denoising step, resulting in $K/B = 7$ denoising steps during the alternating phase. Each drag operation includes $J = 3$ motion supervision steps in Eq. 7. Similar to Shi et al. (2023), we incorporate the Latent-MasaCtrl mechanism (Cao et al., 2023) starting from the 10th U-Net layer to enhance editing performance. We evaluate the runtime and GPU memory usage of GoodDrag with an A100 GPU. For an input image of size 512×512, the LoRA phase takes approximately 10 seconds, while the remaining editing steps require about one minute. The total GPU memory consumption during this process is less than 13GB.

### 5.2  COMPARISON WITH SOTA

As Drag100 provides a more comprehensive and generalizable testbed compared to prior benchmarks (Shi et al., 2023), we present our main evaluation on the Drag100 dataset, with supplementary results on the data of (Shi et al., 2023) presented in the Appendix.

**Qualitative evaluation.** We first compare GoodDrag with DragGAN (Pan et al., 2023) in Figure 7. The proposed method is able to effectively edit the input images, whereas DragGAN suffers from notable artifacts and low fidelity. This superior performance is primarily due to the enhanced generative capabilities of diffusion models compared to GANs, which enables GoodDrag to generalize well across various inputs.

Next, we compare our method with diffusion-based approaches: DragDiffusion (Shi et al., 2023), SDE-Drag (Nie et al., 2023), and DragonDiffusion (Mou et al., 2024a). As shown in Figure 7, DragDiffusion struggles with accurately tracking handle points and often fails to

Table 1: Quantitative evaluation of drag accuracy in terms of DAI (↓) on Drag100.

| Method | $\gamma = 1$ | $\gamma = 5$ | $\gamma = 10$ | $\gamma = 20$ |
|---|---|---|---|---|
| DragDiffusion | 0.148 | 0.144 | 0.130 | 0.115 |
| DragDiffusion* | 0.119 | 0.110 | 0.098 | 0.092 |
| SDE-Drag | 0.157 | 0.144 | 0.129 | 0.114 |
| DragonDiffusion | 0.213 | 0.199 | 0.183 | 0.166 |
| w/o IP | 0.110 | 0.098 | 0.093 | 0.088 |
| w/o AlDD | 0.090 | 0.079 | 0.072 | 0.070 |
| GoodDrag | **0.070** | **0.067** | **0.064** | **0.062** |

Table 2: Quantitative evaluation of image quality in terms of GScore (0 to 10, higher=better) on Drag100. We repeated the experiment 10 times.

| Method | GScore ↑ |
|---|---|
| DragDiffusion | $6.75 \pm 0.10$ |
| SDEDrag | $5.81 \pm 0.19$ |
| DragonDiffusion | $3.05 \pm 0.17$ |
| GoodDrag | $\mathbf{8.04 \pm 0.05}$ |

move semantic content to target locations. While SDE-Drag and DragonDiffusion achieve better point movement, they introduce severe artifacts, resulting in low-fidelity and unrealistic details. In contrast, GoodDrag precisely drags content to specified control points, delivering higher-quality results.

**Quantitative evaluation.** The evaluation in terms of DAI is presented in Table 1, with the patch radius $\gamma$ varying from 1 to 20. A larger $\gamma$ encompass more contextual pixels, offering a broader view of drag accuracy.

As shown in Table 1, GoodDrag consistently outperforms all baseline methods across all $\gamma$ values, indicating superior accuracy in dragging semantic content to target points. Notably, DragDiffusion uses 80 drag operations, while GoodDrag uses 70. With $J = 3$ motion supervision steps per operation (Eq. 7), GoodDrag totals 210 steps, unlike DragDiffusion requires a single step per drag operation. To isolate the impact of more motion supervision steps, we created DragDiffusion*, using 210 drag operations to match GoodDrag. Although this improved the result of DragDiffusion, it still performed worse than GoodDrag, confirming the effectiveness of our approach.

The GScore in Table 2 evaluates the naturalness and fidelity of edited images. Our method achieves an average GScore of 8.04 on the Drag100 dataset, clearly outperforming DragDiffusion, SDE-Drag, and DragonDiffusion.

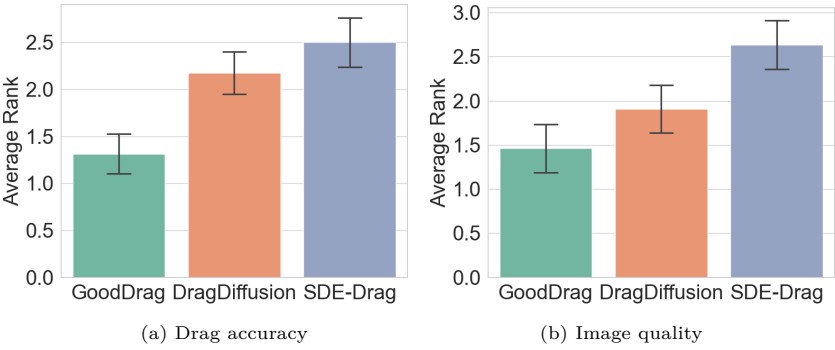

(a) Drag accuracy       (b) Image quality

Figure 8: User study on the drag accuracy (a) and perceptual quality (b) of the edited results. Lower ranks indicate better performance.

**User study.** For a more comprehensive evaluation of the drag editing algorithms, we conduct a user study with 12 images randomly selected from the Drag100 benchmark. Each image is processed by three different methods: DragDiffusion (Shi et al., 2023), SDE-Drag (Nie et al., 2023), and the proposed GoodDrag. Subjects are asked to rank the edited results by each method with the input image as a reference (1 for the best and 3 for the worst). As shown in Figure 8, the study is divided into two parts, with the ranking criteria being the accuracy of the drag editing and the perceptual quality of the results, respectively. We receive responses from 27 participants, and the mean scores and standard deviations are presented in Figure 8. The proposed method is clearly preferred over other methods, suggesting its better capability in achieving precise drag editing (Figure 8(a)) while maintaining high perceptual quality (Figure 8(b)).

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

## A  ALGORITHM

The whole pipeline of GoodDrag is shown in Algorithm 1.

---

**Algorithm 1** Pipeline of GoodDrag

---

**Require:** Input image $z_0$, binary mask for editable region M, handle points $\{\boldsymbol{p}_i\}_{i=1}^l$, target points $\{\boldsymbol{q}_i\}_{i=1}^l$, U-Net $U_\theta$, latent time step $T$, number of drag iterations $K$, number of motion supervision steps per point tracking $J$
**Ensure:** Output image $\hat{z}_0$

1: Finetune $U_\theta$ on $z_0$ with LoRA
2: $z_T \leftarrow$ apply DDIM inversion to $z_0$
3: $z_T^0 \leftarrow z_T$, $\boldsymbol{p}_i^0 \leftarrow \boldsymbol{p}_i$
4: **for** $k$ in $0 : K - 1$ **do**
5: $\quad$ $t = T - \left\lfloor \frac{k}{B} \right\rfloor$
6: $\quad$ $z_{t,0}^k \leftarrow z_t^k$
7: $\quad$ **for** $j$ in $0 : J - 1$ **do**
8: $\quad\quad$ $F(z_{t,j}^k) \leftarrow \mathcal{I}\left(U_\theta(z_{t,j}^k; t)\right)$
9: $\quad\quad$ Update $z_{t,j+1}^k$ using motion supervision as Eq. 7
10: $\quad$ $z_t^{k+1} \leftarrow z_{t,J}^k$
11: $\quad$ Update $\{\boldsymbol{p}_i^{k+1}\}_{i=1}^l$ using points tracking as Eq. 5
12: $\quad$ **if** $(k + 1) \bmod B = 0$ **then**
13: $\quad\quad$ $z_{t-1}^{k+1} \leftarrow$ one step denoising from $z_t^{k+1}$ with Eq. 2
14: **for** $t$ in $T - \frac{K}{B} : 1$ **do**
15: $\quad$ $z_{t-1}^K \leftarrow$ one step denoising from $z_t^K$ with Eq. 2
16: $\hat{z}_0 \leftarrow z_0^K$

---

## B  DDIM INVERSION

The deterministic nature of DDIM allows the transformation of a natural image $z_0$ to its latent variable $z_t$ (the inverse operation of Eq. 2). As suggested in (Song et al., 2020a), the inversion from $z_{t-1}$ to $z_t$ is formulated as:

$$z_t = \sqrt{\alpha_t} \left( \sqrt{\frac{1}{\alpha_t} - 1} - \sqrt{\frac{1}{\alpha_{t-1}} - 1} \right) \cdot \varepsilon_\theta(z_{t-1}, t - 1) + \sqrt{\frac{\alpha_t}{\alpha_{t-1}}} z_{t-1}, \qquad (9)$$

which can be directly derived from Eq. 2, where $\varepsilon_\theta(z_{t-1}, t - 1)$ is used to approximate $\varepsilon_\theta(z_t, t)$. The DDIM inversion is invaluable for image editing applications, where one can first invert $z_0$ to latent space $z_t$, then apply targeted modifications to latent image $z_t$, and finally transform the edited latent image back to the image space by denoising with Eq. 2. This circumvents the difficulties of directly modifying $z_0$, enabling more flexible and practical image editing applications.

## C  RESULTS ON DRAGBENCH

We present the quantitative evaluation on DragBench (Shi et al., 2023) in Table 3 and Table 4. Our method consistently achieves the lowest DAI scores across all $\gamma$ values in Table 3, indicating its superior accuracy in dragging content to target points. Additionally, as shown in Table 4, the edited images from our method demonstrate significantly better GScore, indicating higher fidelity and naturalness compared to other approaches, which further highlights the effectiveness of GoodDrag.

We also provide qualitative evaluations in Figure 9, where our method achieves accurate drag editing while maintaining high fidelity. In contrast, DragonDiffusion struggles to move content precisely to target positions, and both SDE-Drag and DragonDiffusion generate results with noticeable artifacts and unrealistic content.

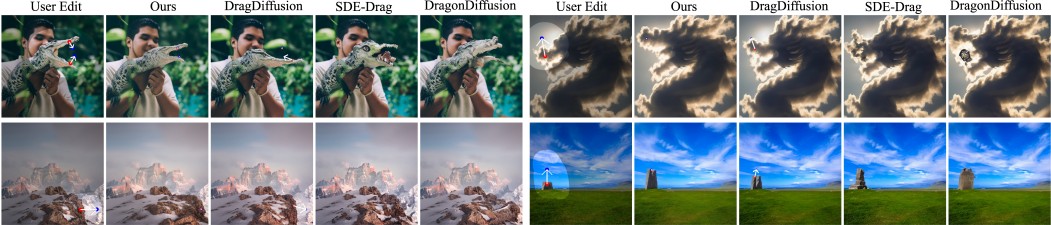

Figure 9: Qualitative comparison on images from other datasets (Shi et al., 2023; Nie et al., 2023). Masks were manually labeled and consistently applied across all methods for fairness. The left column displays results from DragBench dataset, while the right column shows results from SDE-Drag dataset.

Table 3: Quantitative evaluation in terms of DAI ($\downarrow$) on DragBench (Shi et al., 2023).

| Method | $\gamma = 1$ | $\gamma = 5$ | $\gamma = 10$ | $\gamma = 20$ |
|---|---|---|---|---|
| DragDiffusion | 0.1829 | 0.1711 | 0.1618 | 0.1538 |
| SDE-Drag | 0.1796 | 0.1652 | 0.1577 | 0.1499 |
| DragonDiffusion | 0.3108 | 0.2940 | 0.2821 | 0.2692 |
| GoodDrag | **0.1339** | **0.1254** | **0.1210** | **0.1153** |

# D   QUANTITATIVE EVALUATION WITH MD AND IF

For a more comprehensive study, we also adopt the same evaluation metrics as DragDiffusion (Shi et al., 2023), *i.e.*, Mean Distance (MD) and Image Fidelity (IF). The MD metric is defined as the Euclidean distance between the positions of the handle points and the target locations, where the handle points are identified with DIFT (Tang et al., 2023). The IF metric is calculated as 1-LPIPS between the original and edited images.

We conduct comparisons on both the DragBench and Drag100 datasets, as shown in Table 5 and Table 6. The results show that our method achieves significantly better MD values than the baseline methods, demonstrating its effectiveness in accurately dragging content to the desired target locations.

**Limitation of IF.** While the IF score of our method is slightly lower than other approaches, we argue that the IF metric is fundamentally flawed as an evaluation measure for drag editing. IF is defined as 1-LPIPS between the drag-edited image and the input image, meaning it penalizes any changes to the image, even when such changes are necessary to achieve the desired editing. As a result, the metric rewards outputs that are identical or nearly identical to the input image, which contradicts the very purpose of drag editing.

This inherent limitation is clearly demonstrated in Figure 10, where GoodDrag achieves the best visual quality yet receives the worst IF score (0.86), underscoring the inability of IF to accurately evaluate the quality of meaningful drag edits.

In contrast, the proposed GScore metric better correlates with human perception as shown in Figure 10, making it a more reliable and appropriate metric for evaluating drag editing algorithms.

**Runtime of MD.** While MD is effective in measuring drag accuracy, it relies on DIFT Tang et al. (2023) for handle point identification, which significantly increases computational cost and runtime while imposing high demands on GPU resources. Specifically, MD requires an average of 1.8s per image on the Drag100 dataset when using an A100 GPU. This high computational overhead makes it less practical for many users.

In contrast, the proposed DAI metric is much more efficient, requiring only 0.01s per image on average for the Drag100 dataset. Additionally, DAI runs entirely on the CPU, eliminating the need for GPU resources. The efficiency of DAI makes it particularly valuable for drag editing research, where a fast and accessible metric is essential for iterative development and large-scale benchmarking.

Table 4: Quantitative evaluation in terms of GScore (0 to 10, ↑) on DragBench (Shi et al., 2023).

| Method | GScore ↑ |
|---|---|
| DragDiffusion | $6.53 \pm 0.07$ |
| SDEDrag | $5.85 \pm 0.09$ |
| DragonDiffusion | $3.36 \pm 0.18$ |
| Ours | $\mathbf{7.91 \pm 0.04}$ |

Table 5: MD (↓) results on both DragBench and Drag100 datasets.

| Method | Ours | DragDiffusion | SDE-Drag | DragonDiffusion |
|---|---|---|---|---|
| DragBench | **23.40** | 33.50 | 47.84 | 27.04 |
| Drag100 | **23.44** | 37.2 | 74.33 | 28.4 |

## E  ADDITIONAL COMPARISONS WITH MORE BASELINES

We conduct additional comparisons with more baseline approaches, including Drag-Noise (Liu et al., 2024), EasyDrag (Hou et al., 2024), and InstantDrag (Shin et al., 2024). The qualitative results are shown in Figure 11, clearly demonstrating that our proposed GoodDrag achieves superior performance. Specifically, it delivers more accurate drag editing, produces images with significantly higher quality, and minimizes artifacts compared to the baseline methods.

We also present quantitative comparisons in Table 7 where we use DAI to evaluate the proposed GoodDrag against the baseline approaches on Drag100 dataset. Our method consistently outperforms others across all $\gamma$ values, which highlights the robustness and effectiveness of our approach in achieving precise drag edits.

## F  ADDITIONAL DRAGGAN RESULTS

We present a closer visual comparison against DragGAN in Figure 12. Table 8 presents quantitative comparisons between our method and DragGAN using MD, IF, DAI, and GScore. The proposed GoodDrag achieves consistent improvement over DragGAN both qualitatively and quantitatively.

## G  EVALUATION WITHOUT MASK

In our main evaluation, we follow the convention of DragDiffusion and utilize masks by default during the evaluation process.

To provide a more comprehensive analysis, we also compare the performance of different methods without using masks. As shown in Table 9 (with masks) and Table 10 (without masks), the results without masks are generally worse than those with masks, as expected.

Nevertheless, even in the absence of masks, our method consistently outperforms the baseline approaches, demonstrating its robustness and practical effectiveness in real-world scenarios where mask information may not always be provided by user.

Table 6: IF (↑) results on both DragBench and Drag100 datasets.

| Method | Ours | DragDiffusion | SDE-Drag | DragonDiffusion |
|---|---|---|---|---|
| DragBench | 0.87 | 0.88 | **0.91** | 0.90 |
| Drag100 | 0.86 | 0.87 | **0.89** | 0.88 |

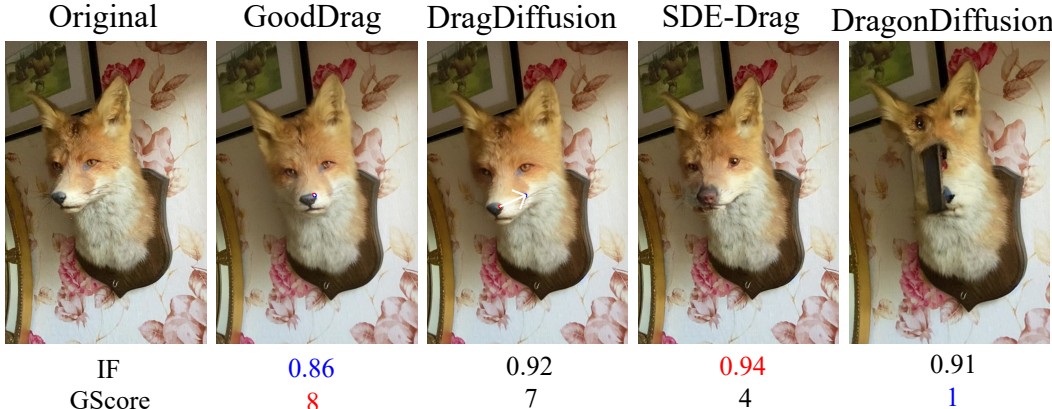

| | Original | GoodDrag | DragDiffusion | SDE-Drag | DragonDiffusion |
|---|---|---|---|---|---|
| IF | | 0.86 | 0.92 | 0.94 | 0.91 |
| GScore | | 8 | 7 | 4 | 1 |

Figure 10: GoodDrag achieves successful drag editing with the best visual quality, but receives the worst IF score (0.86), underscoring the limitation of IF in accurately evaluating meaningful drag edits. In contrast, the proposed GScore metric better correlates with human perception, making it a more reliable and appropriate metric for evaluating drag editing algorithms. Blue numbers indicate the worst scores for each metric, and red ones indicate the best.

Table 7: Quantitative comparison against DragNoise, EasyDrag, and InstantDrag. The evaluation is conducted by measuring the average DAI ($\downarrow$) on Drag100 dataset.

| Method | $\gamma = 1$ | $\gamma = 5$ | $\gamma = 10$ | $\gamma = 20$ |
|---|---|---|---|---|
| DragNoise | 0.209 | 0.191 | 0.169 | 0.146 |
| EasyDrag | 0.201 | 0.191 | 0.169 | 0.142 |
| InstantDrag | 0.173 | 0.152 | 0.128 | 0.108 |
| Ours | **0.070** | **0.067** | **0.064** | **0.062** |

## H EFFECTIVENESS OF AlDD

As introduced in Section 3.2, existing drag editing algorithms often suffer from low fidelity due to the accumulation of perturbations during the drag operations. As shown in Figure 13, the edited result without AlDD exhibits noticeable inconsistencies in the owl's body compared to the original image. In contrast, incorporating AlDD significantly improves the fidelity of the edited result, ensuring that the owl's body remains faithful to the input image.

One might suggest that this fidelity issue could be mitigated by reducing the number of drag operations. However, as illustrated in the second row of Figure 13, while this approach does improve fidelity, it compromises the effectiveness of the drag editing, failing to relocate the content to the desired target locations. This underscores the importance of AlDD in achieving a better balance between fidelity and effective drag editing.

## I EFFECTIVENESS OF INFORMATION-PRESERVING MOTION SUPERVISION

In this section, we evaluate the effectiveness of the information-preserving motion supervision. As shown in Figure 14(b), the model without information-preserving motion supervision suffers from noticeable artifacts as well as dragging failures. In contrast, incorporating the information-preserving strategy effectively mitigates this issue, leading to improved results in Figure 14(d).

The feature distance between the handle point and the original point is shown in Figure 15(b), where the proposed information-preserving motion supervision results in a substantially smaller feature distance (blue curve) compared to the model without this method (orange curve), underscoring its effectiveness in addressing feature drifting issues.

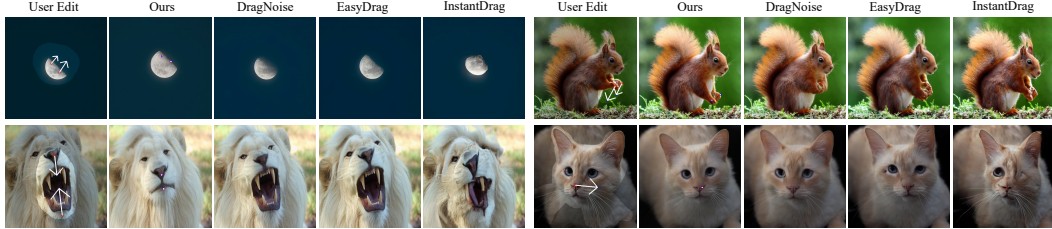

Figure 11: Qualitative comparison with DragNoise (Liu et al., 2024), EasyDrag (Hou et al., 2024), and InstantDrag (Shin et al., 2024).

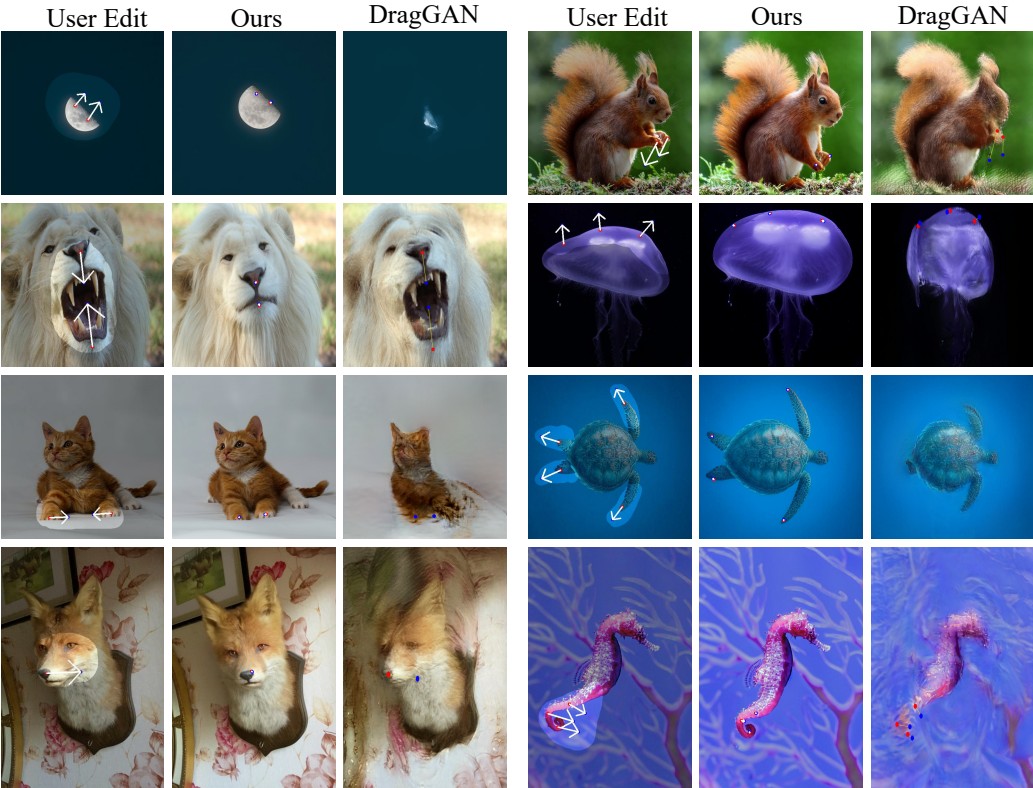

Figure 12: Closer visual comparison with DragGAN.

Furthermore, the information-preserving motion supervision also facilitates more accurate point tracking in Eq. 5. In Figure 15(a), we show the feature distance map between the original point $\boldsymbol{p}_i^0$ and the neighborhood of the current handle point $\Omega(\boldsymbol{p}_i^k, r_2)$. The heatmap with the information-preserving strategy is more concentrated with higher variance, thereby enabling more precise localization of the handle point. In contrast, the heatmap without this strategy is more diffused with lower variance.

Notably, adopting this information-preserving strategy presents challenges in the optimization of motion supervision due to the inherently larger feature distance in Eq. 6 compared to Eq. 3. This increased complexity can impede the movement of the handle point, as shown in Figure 14(c), where the cat's face remains stationary. To overcome this issue, we employ multiple motion supervision steps within a single drag operation. As depicted in Figure 14(d), this approach effectively resolves the above issue, enabling the cat's face dragged to the desired orientation.

Table 8: Quantitative comparison against DragGAN. As DragGAN requires fine-tuning the GAN generator for each input, resulting much slower speed, we only conduct the evaluation on a subset of Drag100 (the six images in Figure 12).

| Metrics | Ours | SDE-Drag | DragDiffusion | DragGAN |
|---|---|---|---|---|
| MD ($\downarrow$) | **15.83** | 67.92 | 57.28 | 73.00 |
| IF ($\uparrow$) | 0.85 | 0.81 | **0.89** | 0.79 |
| DAI ($\gamma = 1$) ($\downarrow$) | **0.078** | 0.156 | 0.189 | 0.196 |
| DAI ($\gamma = 5$) ($\downarrow$) | **0.103** | 0.150 | 0.194 | 0.201 |
| DAI ($\gamma = 10$) ($\downarrow$) | **0.097** | 0.146 | 0.196 | 0.202 |
| DAI ($\gamma = 20$) ($\downarrow$) | **0.070** | 0.129 | 0.178 | 0.187 |
| GScore ($\uparrow$) | **8.10$\pm$0.12** | 7.03$\pm$0.23 | 5.65$\pm$0.35 | 2.12$\pm$0.42 |

Table 9: MD ($\downarrow$) and DAI ($\downarrow$) on DragBench with mask.

| DragBench | Ours | DragDiffusion | SDE-Drag | DragonDiffusion |
|---|---|---|---|---|
| MD | **23.40** | 33.50 | 47.84 | 27.04 |
| DAI ($\gamma = 1$) | **0.1339** | 0.1829 | 0.1796 | 0.3108 |
| DAI ($\gamma = 5$) | **0.1254** | 0.1711 | 0.1652 | 0.2940 |
| DAI ($\gamma = 10$) | **0.1210** | 0.1618 | 0.1577 | 0.2821 |
| DAI ($\gamma = 20$) | **0.1153** | 0.1538 | 0.1499 | 0.2692 |

## J EFFECTIVENESS OF GSCORE AGAINST NR-IQA

We compare various image quality assessment metrics, including TReS (Golestaneh et al., 2022), MUSIQ (Ke et al., 2021), TOPIQ (Chen et al., 2023), and our proposed GScore, in terms of their alignment with human visual perception. We utilize the image quality rankings from the user study in Section 5.2 and measure the correlation between these human rankings and the rankings produced by each metric.

Specifically, for the set of $N_s = 12$ images used in the user study, each image is processed by $N_m = 3$ different methods. For the $i$-th image, the human-assigned rankings for its $N_m$ results are denoted as $\{U_{ij}\}_{j=1}^{N_m}$, where $U_{ij}$ represents the rank assigned to the result of the $j$-th method. The rankings produced by an assessment metric for the same edited results are denoted as $\{R_{ij}\}_{j=1}^{N_m}$.

The correlation between a metric and the human judgment is defined as:

$$\rho = \frac{1}{N_s} \sum_{i=1}^{N_s} \rho_i, \tag{10}$$

where $\rho_i$ is the Spearman's rank correlation coefficient (Gauthier, 2001) for the $i$-th image, calculated as:

$$\rho_i = 1 - \frac{6 \sum_{j=1}^{N_m} (U_{ij} - R_{ij})^2}{N_m(N_m^2 - 1)}. \tag{11}$$

The average correlations are presented in Table 11. While TReS, MUSIQ, and TOPIQ exhibit low (or even negative) correlations, GScore demonstrates a much higher correlation with the human visual system, indicating the effectiveness of GScore for assessing the perceptual quality of drag editing results.

## K GSCORE EXAMPLE

We provide a GScore example in Figure 16.

Table 10: MD (↓) and DAI (↓) on DragBench without mask.

| DragBench | Ours | DragDiffusion | SDE-Drag | DragonDiffusion |
|---|---|---|---|---|
| MD | **23.00** | 36.83 | 48.44 | 25.12 |
| DAI ($\gamma = 1$) | **0.1558** | 0.1972 | 0.1811 | 0.3085 |
| DAI ($\gamma = 5$) | **0.1448** | 0.1914 | 0.1704 | 0.2929 |
| DAI ($\gamma = 10$) | **0.1321** | 0.1781 | 0.1576 | 0.2820 |
| DAI ($\gamma = 20$) | **0.1202** | 0.1654 | 0.1508 | 0.2699 |

Figure 13: Effectiveness of AlDD. In the first row, the result without AlDD shows noticeable inconsistencies in the owl's body compared to the input, while incorporating AlDD effectively addresses this issue. We use 70 drag operations by default. As shown in the second row, reducing the number of drag operations without AlDD improves fidelity but sacrifices the capability in relocating the semantic contents.

## L    ROBUSTNESS ACROSS DIFFERENT BASE MODELS

The proposed GoodDrag framework is compatible with different diffusion base models. While we use Stable Diffusion 1.5 as the default model in this work, we also tested GoodDrag with Stable Diffusion 2.1 and observed minimal difference in performance, which demonstrates the robustness of GoodDrag across different base models. Several examples are provided in Figure 17.

## M    RUNTIME ANALYSIS

Since the proposed Information-Preserving Motion Supervision (IP) involves $J$ motion supervision steps as introduced in Eq. 7, the runtime of GoodDrag is slightly longer than DragDiffusion (71.3s vs. 57.4s) as shown in Table 12.

For a better comparison, we modified DragDiffusion by increasing the number of drag operations to match the number of motion supervision steps used in GoodDrag. While this updated version (referred to as DragDiffusion*) requires a longer runtime, it still underperforms compared to GoodDrag as shown in Table 12, highlighting the advantages of our approach.

Additionally, we tested a simplified version of our model without the IP component, relying solely on the proposed AlDD strategy. This variant (w/o IP) is significantly faster than DragDiffusion (32.1s vs. 57.4s) while still achieving better performance than DragDiffusion. These results further demonstrate the efficiency and efficacy of the proposed algorithm.

## N    RELATIONSHIP WITH DRAGONDIFFUSION AND DIFFEDITOR

DragonDiffusion (Mou et al., 2024a) and DiffEditor (Mou et al., 2024b) are two related works that also involve image editing within the denoising diffusion process. Nevertheless, they are fundamentally different from GoodDrag and the proposed AlDD in both theoretical foundations and practical implementation.

(a) User Edit      (b) w/o IP      (c) w/ IP (Once)      (d) w/ IP

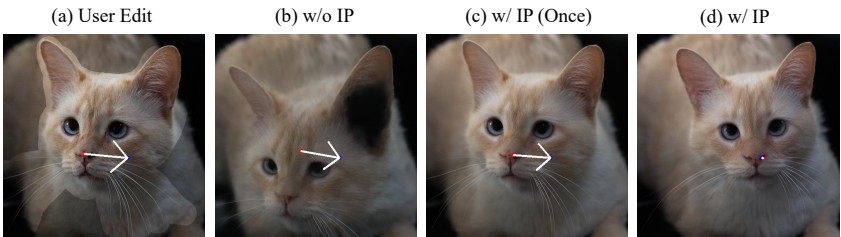

Figure 14: The results of different processing conditions on the subject: (a) User Edit, (b) without the proposed information-preserving motion supervision (IP), (c) with IP applied once, and (d) with IP applied optimally. Without IP, noticeable artifacts and dragging failures occur, as shown in (b). Direct application of IP once is less effective, leading to inferior results as in (c). Employing multiple IP steps within a single drag operation, as optimized in (d), significantly improves the outcome by addressing these issues.

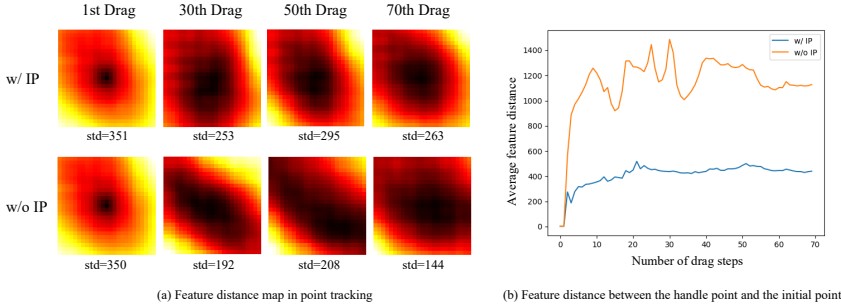

(a) Feature distance map in point tracking      (b) Feature distance between the handle point and the initial point

Figure 15: (a) shows the feature distance map from Eq. 5 at different drag steps. More specifically, these heatmaps represent the feature distances between the original point $\boldsymbol{p}_i^0$ and the neighborhood of the current handle point $\Omega(\boldsymbol{p}_i^k, r_2)$. The standard deviation (std) of the distances in each heatmap is provided below, where a small std indicates a diffused heatmap with indistinctive feature distances, and a large std indicates a more concentrated heatmap, resulting in generally more accurate localization of the smallest distance in Eq. 5. (b) shows the feature distance between the handle point and the original point with the increase of drag steps. The distance with the proposed information-preserving motion supervision (IP) is much smaller than that without IP, demonstrating its effectiveness in dealing with the feature drifting issue.

From a theoretical perspective, DragonDiffusion and DiffEditor rely on a mechanism analogous to classifier guidance (Dhariwal & Nichol, 2021). In these methods, the diffusion process remains probabilistically grounded, and each step is guided by combining the unconditional gradient and the conditional likelihood term. Mathematically, this is expressed as:

$$\nabla_{\mathbf{x}_t} \log q(\mathbf{x}_t \mid \mathbf{y}) = \nabla_{\mathbf{x}_t} \log q(\mathbf{x}_t) + \nabla_{\mathbf{x}_t} \log q(\mathbf{y} \mid \mathbf{x}_t), \tag{12}$$

where the guidance operates within the probabilistic framework of diffusion (see Eq. 8 of DragonDiffusion (Mou et al., 2024a) and Eq. 3 of DiffEditor (Mou et al., 2024b)). From a practical perspective, this results in the editing and denoising processes being intertwined and inseparable.

In contrast, GoodDrag follows a fundamentally different paradigm, similar to DragDiffusion (Shi et al., 2023), where the drag editing operations and the denoising diffusion process are decoupled. AlDD distributes drag operations strategically across multiple diffusion steps but is not constrained by the probabilistic formulation of classifier guidance, which represents a significant departure from existing methods. This separation allows AlDD to introduce flexibility in drag editing, which is not feasible with methods like DragonDiffusion and DiffEditor, and effectively improves the results. As shown in Section 5.2 and Appendix C and D, GoodDrag achieves significantly better performance than DragonDiffusion across

Table 11: Correlations between various image quality assessment metrics and human visual perception.

|  | TReS | MUSIQ | TOPIQ | GScore |
|---|---|---|---|---|
| $\rho \uparrow$ | 0.250 | -0.125 | 0.083 | **0.54** |

Prompt: Conduct a detailed evaluation of three modified images, labeled 'A', 'B', 'C' and 'D', in comparison to an original image (Image 1). Image 1 serves as the baseline and will not be evaluated. Focus on assessing the quality of 'A' (Image 2), 'B' (Image 3), 'C' (Image 4), and 'D' (Image 5), particularly in terms of their naturalness and the presence or absence of artifacts. Examine how well each algorithm preserves the integrity of the original image while introducing modifications. Look for any signs of distortions, unnatural colors, pixelation, or other visual inconsistencies. Rate each image on a scale from 1 to 10, where 10 represents excellent quality with seamless modifications, and 1 indicates poor quality with significant and noticeable artifacts. Provide a comprehensive analysis for each rating, highlighting specific aspects of the image that influenced your evaluation. Answers must be in English.

Image 1

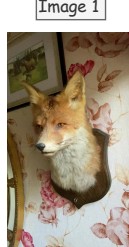

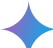

**Image 2 (A): 8/10**

Image 2 (A) is a successful modification of the original image. The fox's fur appears natural and well-textured, and the overall color balance is pleasing. The modification is subtle yet noticeable, adding a touch of whimsy to the original image without detracting from its realism. The only minor issue is a slight distortion of the fox's left ear, which appears slightly elongated.

**Image 3 (B): 7/10**

Image 3 (B) is another well-done modification. The fox's fur is perhaps even more realistic than in the original image, and the colors are vibrant and lifelike. The modification is more pronounced than in Image 2 (A), but it still maintains a sense of naturalness. The only noticeable artifact is a slight pixelation around the edges of the fox's ears.

**Image 4 (C): 4/10**

Image 4 (C) is a more drastic modification of the original image, and it is less successful than the previous two. The fox's fur appears unnatural and almost plastic-like, and the colors are too saturated. The modification is also quite noticeable, and it detracts from the realism of the original image. There are also some noticeable distortions around the fox's eyes and mouth.

**Image 5 (D): 1/10**

Image 5 (D) is a complete failure. The fox's face is completely distorted, and the colors are garish and unnatural. The modification is so extreme that it is difficult to even recognize the original image. There are also numerous artifacts, including pixelation, blurring, and unnatural-looking shadows.

In conclusion, Image 2 (A) is the most successful modification, followed by Image 3 (B). Image 4 (C) is a less successful modification, and Image 5 (D) is a complete failure. The main factors that contributed to the success of Image 2 (A) and Image 3 (B) were the naturalness of the fur and the pleasing color balance. The main factors that contributed to the failure of Image 4 (C) and Image 5 (D) were the unnatural appearance of the fur, the distorted shapes, and the garish colors.

Figure 16: An example from GScore: Images A, B, C, and D in Figure 7, shown in the last row of the left column, represent GoodDrag, DragDiffusion, SDE-Drag, and DragonDiffusion, respectively. For each prompt, we input the original image along with the comparison images.

multiple benchmarks, both quantitatively and qualitatively. These results underline the practical advantages of AlDD and the distinctiveness of GoodDrag's approach.

## O  LIMITATIONS

Similar to existing diffusion-based methods, such as DragDiffusion, the proposed GoodDrag relies on DDIM inversion for effective drag editing. However, DDIM inversion may face challenges in complex scenarios, as illustrated in Figure 18, where the reconstruction of the

| User Edit | GoodDrag-SD1.5 | GoodDrag-SD2.1 | User Edit | GoodDrag-SD1.5 | GoodDrag-SD2.1 |

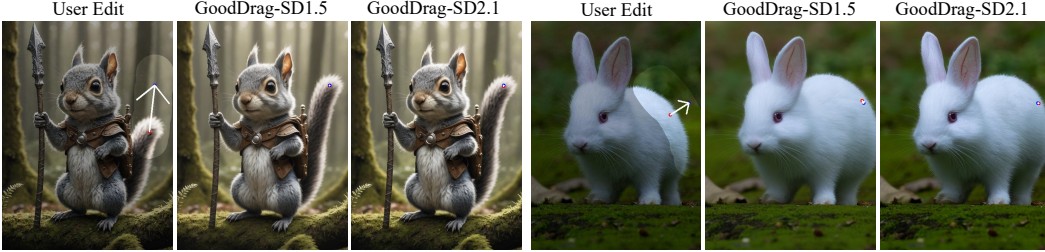

Figure 17: The proposed GoodDrag demonstrates consistent performance with different diffusion base models.

Table 12: Comparing the runtime of GoodDrag and DragDiffusion.

|  | GoodDrag | DragDiffusion* | w/o IP | DragDiffusion |
|---|---|---|---|---|
| Number of Motion Supervision steps | 210 | 210 | 70 | 80 |
| Runtime (s) | 71.3 | 80.1 | **32.1** | 57.4 |
| DAI ($\gamma = 1$) | **0.070** | 0.119 | 0.110 | 0.148 |
| DAI ($\gamma = 5$) | **0.067** | 0.110 | 0.098 | 0.144 |
| DAI ($\gamma = 10$) | **0.064** | 0.098 | 0.093 | 0.130 |
| DAI ($\gamma = 20$) | **0.062** | 0.092 | 0.088 | 0.115 |

inversed image (Figure 18(b)) appears blurred and many fine details are lost. Consequently, the edited result of GoodDrag also suffers from these artifacts as shown in Figure 18(d). In future work, we aim to explore more robust and effective diffusion inversion techniques for better drag editing performance.

| (a) Input Image | (b) Reconstruction from DDIM Inversion | (c) User Edit | (d) GoodDrag |

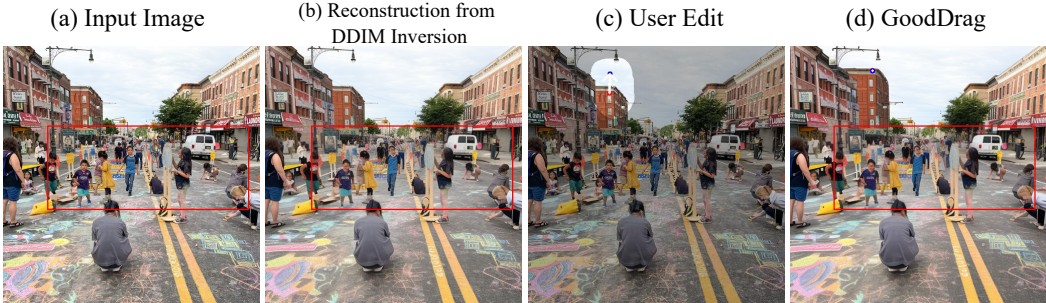

Figure 18: Limitation. The proposed GoodDrag relies on DDIM inversion, which may struggle with complex images, where many fine details of the original input cannot be clearly restored from the inversed image (b).

## P    ETHICS STATEMENT

GoodDrag enhances image editing capabilities, benefiting creative industries and digital content creation by providing more precise and reliable tools. However, its advanced manipulation features could be misused to create misleading or deceptive content, such as deepfakes. While we release the source code and dataset to support research and development, we encourage users to adhere to ethical standards and applicable regulations to prevent misuse.

## Q    CONCLUDING REMARKS

In this work, we introduce GoodDrag, a method that enhances the stability and quality of drag editing. Leveraging our AlDD framework, we effectively mitigate distortions and enhance image fidelity by distributing drag operations across multiple diffusion denoising steps. In addition, we introduce information-preserving motion supervision to tackle the feature drifting issue, thereby reducing artifacts and enabling more precise control over handle points. Furthermore, we present the Drag100 dataset and two dedicated evaluation metrics, DAI and GScore, to facilitate a more comprehensive benchmarking of the progress in drag editing. The simplicity and efficacy of GoodDrag establish a strong baseline for the development of more sophisticated drag editing algorithms. Future directions include exploring the integration of GoodDrag with other image editing tasks and extending its capabilities to video editing scenarios.

