# OpenReview forum: "GoodDrag: Towards Good Practices for Drag Editing with Diffusion Models"
_ICLR.cc/2025/Conference — ICLR 2025 Poster_

### Official Review · Reviewer_pphp · 2024-10-16

**Soundness:** 2
**Presentation:** 3
**Contribution:** 3
**Rating:** 6
**Confidence:** 4

**Summary:**

The paper introduces GoodDrag, a novel framework designed to enhance the stability and image quality of drag editing using diffusion models. The main contributions of the paper are as follows:

- Alternating Drag and Denoising (AlDD) Framework: GoodDrag employs the AlDD framework, which alternates between drag and denoising steps within the diffusion process. This approach prevents the accumulation of large perturbations, ensuring more refined and manageable editing results. The AlDD framework is depicted as a significant improvement over existing methods that apply all drag operations at once, followed by denoising, which often leads to excessive perturbations and low fidelity.
- Information-Preserving Motion Supervision: To combat common failures in drag editing related to point control, GoodDrag introduces an information-preserving motion supervision operation. This method maintains the original features of the starting point, ensuring more realistic and precise control over the drag process and reducing artifact generation.
- Benchmarking Enhancements: The paper contributes to the benchmarking of drag editing techniques by introducing a new dataset called Drag100 and developing dedicated quality assessment metrics, namely the Dragging Accuracy Index (DAI) and the Gemini Score (GScore). These metrics are designed to evaluate the quality of drag editing more effectively, utilizing Large Multimodal Models for a more accurate assessment.

**Strengths:**

1. **Introduction of the Drag100 Dataset**: The creation of the Drag100 dataset is a significant advancement for the field of drag editing. This new dataset provides a well-defined and controlled environment for testing and benchmarking drag editing algorithms. It includes a variety of image types and editing scenarios, which ensures comprehensive testing across different contexts and challenges. This diversity helps in evaluating the robustness and versatility of new drag editing methods under varied conditions.

2. **Development of Novel Evaluation Metrics**: The paper introduces two new evaluation metrics, the Dragging Accuracy Index (DAI) and the Gemini Score (GScore). These metrics are designed to provide a more precise and relevant assessment of drag editing quality. DAI focuses on the accuracy of content transfer to target locations, while GScore evaluates the naturalness and fidelity of the edited images using Large Multimodal Models. These metrics contribute to a standardized approach for assessing image editing quality, promoting more objective and reproducible research outcomes in the field.

3. **Well-Structured Presentation and Comprehensive Validation**: The paper excels in its presentation, clearly outlining the methodologies, experiments, and results.

**Weaknesses:**

1. **Questionable Novelty of AlDD**: The paper claims that existing methods typically conduct all drag operations at once and then attempt to correct accumulated perturbations subsequently. However, methods like DragonDiffusion and DiffEditor have previously explored applying motion supervision across multiple diffusion time steps. This raises questions about the purported novelty of AlDD, as the concept of distributing editing operations across multiple steps is not entirely new.

2. **Lack of Running Time Analysis**: The paper does not provide a comparative analysis of the running times of GoodDrag versus other state-of-the-art methods. Including this information would help evaluate the practical applicability of GoodDrag, especially in real-time or resource-constrained scenarios.

3. **Evaluation Metrics - Mean Distance and LPIPS**: DragDiffusion has already introduced Mean Distance and LPIPS as evaluation metrics for assessing drag editing quality. The paper would benefit from reporting these metrics alongside the newly proposed DAI and GScore to provide a comprehensive evaluation framework that can be directly compared with existing methods.

4. **Comparison with More Recent Methods**: The paper lacks comparisons with more recent advancements in drag editing, such as DiffEditor, InstantDrag and LightningDrag. Including these methods in comparative evaluations would strengthen the paper's relevance and demonstrate the efficacy of GoodDrag in the context of the latest research developments.

5. **Use of DragBench and Handling of Masks**: The paper criticizes some datasets for not consistently providing masks, yet a drag-editing model should not alway relies on user-provided masks. And in the original DragGAN formulation, the mask is optional, highlighting a discrepancy in the paper's critique. A more robust approach would be to evaluate GoodDrag's performance on both DragDiffusion and SDEDrag's DragBench (both with and without masks), thereby demonstrating its versatility and alignment with user-friendly design principles seen in earlier methods.

**Questions:**

**Clarification on Tracking Steps in the Presence of Inversion Strength**: Does the GoodDrag framework continue to utilize point tracking steps within the Alternating Drag and Denoising (AlDD) framework? If so, considering the inversion strength is set to 0.7, which corresponds to a significant amount of noise reduction and thus a deviation from the original image features, how does the system ensure accurate matching and tracking of points? Specifically, since diffusion-based methods can maintain correspondence primarily when the diffusion time step is small, how does GoodDrag handle scenarios where the dragged points do not align with their intended targets due to these potential mismatches in feature correspondence?

---

> ### Author Response · Authors · 2024-11-27
> **Reponse to Reviewer pphp**
>
> We thank the reviewer for the valuable comments on this paper and address the raised issues here and in the revised manuscript.
>
> **Q1: DragonDiffusion and DiffEditor**
>
> **A1**: Thank you for highlighting the related works, DragonDiffusion and DiffEditor. While these methods involve image editing within the denoising diffusion process, they are fundamentally different from GoodDrag and the proposed AlDD in both theoretical foundations and practical implementation.
>
> From a theoretical perspective, DragonDiffusion and DiffEditor rely on a mechanism analogous to classifier guidance. In these methods, the diffusion process remains probabilistically grounded, and each step is guided by combining the unconditional gradient and the conditional likelihood term. Mathematically, this is expressed as:
>
> $\nabla_{x_t} \log q(x_t \mid y) = \nabla_{x_t} \log q(x_t) + \nabla_{x_t} \log q(y \mid x_t)$
>
> where the guidance operates within the probabilistic framework of diffusion (see Eq. 8 of DragonDiffusion and Eq. 3 of DiffEditor). From a practical perspective, this results in the editing and denoising processes being intertwined and inseparable.
>
> In contrast, GoodDrag follows a fundamentally different paradigm, similar to DragDiffusion, where the drag editing operations and the denoising diffusion process are decoupled. AlDD distributes drag operations strategically across multiple diffusion steps but is not constrained by the probabilistic formulation of classifier guidance, which represents a significant departure from existing methods. This separation allows AlDD to introduce flexibility in drag editing, which is not feasible with methods like DragonDiffusion and DiffEditor, and effectively improves the results. As shown in Section 5.2 and Appendix B-C, GoodDrag achieves significantly better performance than DragonDiffusion across multiple benchmarks, both quantitatively and qualitatively. These results underline the practical advantages of AlDD and the distinctiveness of GoodDrag’s approach.
>
> **Q2: Runtime analysis.**
>
> **A2**: We include runtime analysis in Appendix L of our paper. Since the proposed Information-Preserving Motion Supervision (IP) involves $J$ motion supervision steps as introduced in Eq. 7, the runtime of GoodDrag is slightly longer than DragDiffusion (71.3s vs. 57.4s) as shown in the table below.
> For a better comparison, we modified DragDiffusion by increasing the number of drag operations to match the number of motion supervision steps used in GoodDrag. While this updated version (referred to as DragDiffusion*) requires a longer runtime, it still underperforms compared to GoodDrag as shown in the table below, highlighting the advantages of our approach.
>
> Additionally, we tested a simplified version of our model without the IP component, relying solely on the proposed AlDD strategy. This variant (w/o IP) is significantly faster than DragDiffusion (32.1s vs. 57.4s) while still achieving better performance than DragDiffusion. These results further demonstrate the efficiency and efficacy of the proposed algorithm.
>
> |                                       | **Ours** | **DragDiffusion*** | **Ours w/o IP** | **DragDiffusion** |
> |-----------------------------------------------------|--------------|--------------------|------------|-------------------|
> | Number of Motion Supervision Steps                 | 210          | 210                | 70         | 80                |
> | Runtime (s)                                        | 71.3         | 80.1               | **32.1**   | 57.4              |
> | DAI ($\gamma=1$)                               | **0.070**    | 0.119              | 0.110      | 0.148             |
> | DAI ($\gamma=5$)                                | **0.067**    | 0.110              | 0.098      | 0.144             |
> | DAI ($\gamma=10$)                               | **0.064**    | 0.098              | 0.093      | 0.130             |
> | DAI ($\gamma=20$)                               | **0.062**    | 0.092              | 0.088      | 0.115             |

---

> > ### Author Response · Authors · 2024-11-27
> > **Reponse to Reviewer pphp**
> >
> > **Q3: Evaluation with Mean Distance and LPIPS.**
> >
> > **A3**: As suggested, we provide the MD and IF (1-LPIPS) results in Table 5 and 6 of the revised Appendix C, and also include them below for easier reference. Our method achieves better MD values than the baseline methods, demonstrating its effectiveness in drag editing. Note that while MD can well measure drag accuracy, it is considerably slower than the proposed DAI as detailed in Appendix C.
> >
> >    | MD ($\downarrow$)       	| Ours  | DragDiffusion | SDEDrag | DragonDiffusion |
> >    |------------------|-------|---------------|---------|-----------------|
> >    | DragBench    	| **23.40** | 33.50     	| 47.84   | 27.04       	|
> >    | Drag100      	| **23.44** | 37.20     	| 74.33   | 28.40       	|
> >
> >  | IF ($\uparrow$)       	| Ours | DragDiffusion | SDEDrag | DragonDiffusion |
> >    |------------------|------|---------------|---------|-----------------|
> >    | DragBench    	| 0.87 | 0.88      	| **0.91**	| 0.90        	|
> >    | Drag100      	| 0.86 | 0.87      	| **0.89**	| 0.88        	|
> >
> > While the IF score of our method is slightly lower than other approaches, we argue that the IF metric is fundamentally flawed as an evaluation measure for drag editing approaches. IF is defined as 1-LPIPS between the drag-edited image and the input image, meaning it penalizes any changes to the image, even when such changes are necessary to achieve the desired editing. As a result, the metric rewards outputs that are identical or nearly identical to the input image, which contradicts the very purpose of drag editing.
> >
> > This inherent limitation is clearly demonstrated in Figure 11 of the revised Appendix C, where an example with superior visual quality receives the worst IF score, underscoring its failure to reflect meaningful drag edits. In contrast, the proposed GScore metric better correlates with human perception, making it a more reliable and appropriate metric for evaluating drag editing algorithms. Please refer to Appendix C for more details.
> >
> > **Q4: Comparison with more recent methods**
> >
> > **A4**: We provide comparisons with additional baselines in Appendix D and Table 7 of the revised paper, including InstantDrag, DragNoise, and EasyDrag. The proposed GoodDrag compares favorably against these methods, showcasing its effectiveness.
> >
> > Regarding DiffEditor, we were unable to find its implementation code. After reviewing the associated repository: https://github.com/MC-E/DragonDiffusion/tree/master, we only found the code for DragonDiffusion, which we have already included in our comparisons in Table 1–6 of the paper.
> >
> > As for LightningDrag, it is a concurrent work that has not yet been peer-reviewed or accepted, as it is currently only available as an ArXiv preprint that was released after our GoodDrag. Despite this, we have acknowledged and cited all these relevant works in our main paper to ensure comprehensive coverage of related research.
> >
> > **Q5: Use of DragBench and handling of masks**
> >
> > **A5**: In our main evaluation, we followed the convention of DragDiffusion and utilize masks by default during the evaluation process. As per the reviewer’s suggestion, we provide additional results on DragBench both with and without masks in Appendix F of the revised paper.
> >
> > As shown in Table 9 (with masks) and Table 10 (without masks) of our paper, the results without masks are generally worse than those with masks, as expected. Nevertheless, even in the absence of masks, our method consistently outperforms the baseline approaches, demonstrating its robustness and practical effectiveness in real-world scenarios where mask information may not always be provided by user.
> >
> >
> > **Q6: Clarification on Tracking Steps in the Presence of Inversion Strength**
> >
> > **A6**: Yes, GoodDrag utilizes point tracking within the Alternating Drag and Denoising (AlDD) framework, as detailed in Section 3.2 and Appendix A.
> >
> > Point tracking is achieved, as described in Eq. 5, by identifying the point $\boldsymbol{p}$ in the current edited image $z_t^{k+1}$ that is most similar to the source point $\boldsymbol{p}_i^0$ in the original image $z_t^0$. Note that we ensure that the current image $z_t^{k+1}$ and the original image $z_t^0$ are aligned at the same timestep t. This alignment ensures that both images reside in the same latent space, preserving consistent feature representations. By maintaining this consistency, mismatched issues are effectively avoided, allowing our point tracking mechanism to reliably find corresponding points.

---

> > ### Comment · Reviewer_pphp · 2024-11-28
> >
> > Thank you for your detailed reply, which has largely addressed my concerns. However, I still have one remaining concern regarding the comparison to DiffEditor. While I acknowledge that GoodDrag demonstrates superior performance compared to DiffEditor, both methods alternate between dragging and denoising during the diffusion process. The main distinction lies in the fact that DiffEditor doesn't implement track-and-drag like GoodDrag does, or it simply sets the dragging step to 1. Given these similarities, it's difficult to consider 'AIDD' as an entirely novel contribution. I will increase the rating if this concern can be solved.
> >
> > An optional improvement is to discuss FastDrag, RegionDrag in related work.

---

> ### Author Response · Authors · 2024-11-29
> **Further Clarification**
>
> We thank the reviewer for the detailed follow-up and for raising this important concern. Here, we provide additional clarification on the differences between DiffEditor and GoodDrag, specifically addressing the novelty of AlDD.
>
> Contrary to the claim that DiffEditor alternates between dragging and denoising during the diffusion process, the drag and denoising steps in DiffEditor are performed together, not separately. For further reference, please see Algorithm 1 on Page 5 of the [DiffEditor](https://arxiv.org/pdf/2402.02583) paper. The editing operation in DiffEditor is executed via Line 17 in Algorithm 1, defined as:
>
> $\mathbf{z} _ {t-1}=\mathbf{m} _ {edit}\cdot \mathcal{F}(\mathbf{z} _ t;\eta_1(t))+(1-\mathbf{m}_{edit})\cdot \mathcal{F}(\mathbf{z}_t;\eta_2(t))$
>
> To more clearly see why drag and denoising are intertwined in DiffEditor, consider the expanded form of the first term $\mathcal{F}(\mathbf{z}_t;\eta_1(t))$ using Eq. 2 of [DiffEditor](https://arxiv.org/pdf/2402.02583) (the second term $\mathcal{F}(\mathbf{z}_t;\eta_2(t))$ is similar and omitted for clarity):
>
> $\mathcal{F}(\mathbf{z} _ t;\eta_1(t))=\sqrt{\alpha _ {t-1}} \frac{\mathbf{z} _ {t}-\sqrt{1-\alpha _ {t}} \tilde{{\epsilon}} _ {\theta}^{t}\left(\mathbf{z} _ {t}\right)}{\sqrt{\alpha _ {t}}} + \sqrt{1-\alpha _ {t-1}-\eta_1(t)^{2}} \cdot \tilde{{\epsilon}} _ {\theta}^{t}\left(\mathbf{z}_{t}\right) + \eta_1(t) {\epsilon}$
>
> With Eq. 4 from [DiffEditor](https://arxiv.org/pdf/2402.02583), we further expand the formula as:
>
> $\mathcal{F}(\mathbf{z} _ t;\eta_1(t))=\sqrt{\alpha _ {t-1}} \frac{\mathbf{z} _ {t}-\sqrt{1-\alpha _ {t}} \left[\epsilon _ {\theta}^{t}\left(\mathbf{z} _ {t}\right)+\eta\cdot\nabla _ {\mathbf{z} _ t} \mathcal{E}(\mathbf{z} _ t, \mathbf{y})\right]}{\sqrt{\alpha _ {t}}} + \sqrt{1-\alpha _ {t-1}-\eta _ 1(t)^{2}} \cdot \left[\epsilon _ {\theta}^{t}\left(\mathbf{z} _ {t}\right) +\eta\cdot\nabla _ {\mathbf{z} _ t} \mathcal{E}(\mathbf{z} _ t, \mathbf{y})\right] + \eta _ 1(t) \epsilon$
>
> Here, ${\epsilon} _ {\theta}^{t}\left(\mathbf{z} _ {t}\right)$ represents the denoising signal, while $\nabla_{\mathbf{z}_t} \mathcal{E}(\mathbf{z}_t, \mathbf{y})$ represents the drag signal. These two signals are fused and inherently intertwined in DiffEditor’s editing process and cannot be easily decoupled.
>
> This design is deeply rooted in the probabilistic framework of classifier guidance (Eq. 12 on Page 8 of ["Diffusion Models Beat GANs on Image Synthesis"](https://arxiv.org/pdf/2105.05233) or Eq. 3 on Page 3 of [DiffEditor](https://arxiv.org/pdf/2402.02583)), which mandates that the denoising and guidance signals be applied together.
>
> It’s worth noting that the $t$%$2==0$ on Line 13 of DiffEditor’s Algorithm 1 might be mistaken for alternating drag and denoising. However, as explained on Page 6 of DiffEditor, it is to reduce the number of guidance timesteps, which we quote:
>
> “Due to the guidance enhancement from regional guidance and time travel, we can achieve editing with fewer guidance time steps, i.e., we introduce gradient guidance every two time steps in sampling.”
>
> This does not decouple the drag and denoising operations in $\mathcal{F}(\mathbf{z}_t;\eta_1(t))$.
>
> In contrast, GoodDrag, through the AlDD framework, explicitly separates the dragging and denoising processes. This separation is not a superficial design choice but rather a foundational distinction that allows AlDD to strategically alternate between image editing (motion supervision and point tracking) and artifact reduction (denoising). This independence offers greater flexibility and precision in managing drag operations, which is not achievable in DiffEditor’s tightly coupled probabilistic framework.
>
> **Discussion on Related Work.** We have cited both FastDrag and RegionDrag in the related work of our paper. While the deadline for updating the PDF has passed, we will include a more detailed discussion of these works in the final version.
>
> We hope this clarification addresses the remaining concern, and we thank the reviewer for the opportunity to elaborate further.

---

> > ### Author Response · Authors · 2024-11-30
> >
> > Dear Reviewer pphp,
> >
> > Thank you for taking the time to review our paper and rebuttal. We greatly appreciate your updated feedback and rating.
> >
> > Your insights are invaluable, and we truly value your input in helping to refine our work. Thank you once again for your time and constructive feedback.

---

### Official Review · Reviewer_k1h1 · 2024-10-24

**Soundness:** 3
**Presentation:** 3
**Contribution:** 3
**Rating:** 6
**Confidence:** 4

**Summary:**

This paper introduces GoodDrag, a diffusion-based approach designed to enhance the stability and fidelity of image drag editing. The core idea is to alternate between drag and denoising operations within the diffusion process to prevent perturbation accumulation, leading to more stable image generation. To this end, the authors propose the Alternating Drag and Denoising (AIDD) framework along with a new benchmark, Drag100. Experimental results demonstrate the effectiveness of the proposed methods both qualitatively and quantitatively.

**Strengths:**

1. The paper is well-written, and the proposed approach is technically sound.
2. The qualitative results well demonstrate the effectiveness of the proposed framework.

**Weaknesses:**

The main concern is the insufficiency of experimental results:
1. Although the authors proposed two quality assessment metrics, widely used evaluation metrics such as Image Fidelity (IF) and Mean Distance (MD) should also be reported.
2. How does the method perform on other benchmarks, such as DRAGBENCH?
3. What is the impact of using different base models on the results?
4. Limitations and failure cases should be discussed and reported.

**Questions:**

Please refer to the weaknesses.

---

> ### Author Response · Authors · 2024-11-27
> **Reponse to Reviewer k1h1**
>
> We thank the reviewer for the valuable comments on this paper and address the raised issues here and in the revised manuscript.
>
> **Q1: Evaluation with IF and MD**
>
> **A1**: We provide the MD and IF results in Table 5 and 6 of the revised Appendix C, and also include them below for easier reference. Our method achieves better MD values than the baseline methods, demonstrating its effectiveness in drag editing. Note that while MD can well measure drag accuracy, it is considerably slower than the proposed DAI as detailed in Appendix C.
>
>    | MD ($\downarrow$)       	| Ours  | DragDiffusion | SDEDrag | DragonDiffusion |
>    |------------------|-------|---------------|---------|-----------------|
>    | DragBench    	| **23.40** | 33.50     	| 47.84   | 27.04       	|
>    | Drag100      	| **23.44** | 37.20     	| 74.33   | 28.40       	|
>
>  | IF ($\uparrow$)       	| Ours | DragDiffusion | SDEDrag | DragonDiffusion |
>    |------------------|------|---------------|---------|-----------------|
>    | DragBench    	| 0.87 | 0.88      	| **0.91**	| 0.90        	|
>    | Drag100      	| 0.86 | 0.87      	| **0.89**	| 0.88        	|
>
> While the IF score of our method is slightly lower than other approaches, we argue that the IF metric is fundamentally flawed as an evaluation measure for drag editing approaches. IF is defined as 1-LPIPS between the drag-edited image and the input image, meaning it penalizes any changes to the image, even when such changes are necessary to achieve the desired editing. As a result, the metric rewards outputs that are identical or nearly identical to the input image, which contradicts the very purpose of drag editing.
>
> This inherent limitation is clearly demonstrated in Figure 11 of the revised Appendix C, where an example with superior visual quality receives the worst IF score, underscoring its failure to reflect meaningful drag edits. In contrast, the proposed GScore metric better correlates with human perception, making it a more reliable and appropriate metric for evaluating drag editing algorithms. Please refer to Appendix C for more details.
>
> **Q2: Results on DragBench**
>
> **A2**: The results on DragBench are provided in Appendix B and C (Tables 3–6). Our GoodDrag demonstrates favorable performance compared to state-of-the-art methods, consistently achieving superior results across different metrics, including DAI, GScore, and MD.
>
> **Q3: Different base model**
>
> **A3**: The proposed GoodDrag framework is compatible with different diffusion base models. While we use Stable Diffusion 1.5 as the default model in this work, we also tested GoodDrag with Stable Diffusion 2.1 and observed minimal difference in performance, which demonstrates the robustness of GoodDrag across different base models. Please refer to Appendix K for more details.
>
> **Q4: Limitation**
>
> **A4**: Similar to existing diffusion-based methods, such as DragDiffusion, the proposed GoodDrag relies on DDIM inversion for effective drag editing. However, DDIM inversion may face challenges in complex scenarios, where the reconstruction of the inversed image appears blurred and many fine details are lost. Consequently, the edited result of GoodDrag also suffers from these artifacts. Please refer to Appendix N for more details.

---

> ### Author Response · Authors · 2024-11-30
>
> Dear Reviewer k1h1,
>
> As the discussion phase is coming to a close, we wanted to check if our responses have fully addressed your concerns. If there are any remaining issues that need clarification, please let us know—we’d be happy to provide additional details.
>
> Thank you!

---

> ### Author Response · Authors · 2024-12-01
> **Follow-Up on Rebuttal Feedback**
>
> Dear Reviewer k1h1,
>
> Thank you for your valuable and constructive feedback on our submission. Your comments have been instrumental in improving the quality of our manuscript.
>
> As the discussion deadline approaches, we want to ensure that all your concerns have been fully addressed. Please feel free to reach out if you require further clarifications or additional analyses.
>
> To address your points:
> - Results for DragBench with MD and IF are in Appendices B and C, with comparisons for DAI, GScore, MD, and IF detailed in Tables 1–6.
>
> - Different base models are discussed in Appendix K.
>
> - A new section has been added in Appendix O to discuss the limitations of our work.
>
> We hope these updates address your concerns and reflect the improvements made to the paper. If you have had a chance to review our rebuttal and revisions, we would greatly appreciate your feedback or confirmation on whether the updated manuscript meets your expectations.
>
> Additionally, we hope you might consider raising your score based on the revisions, as we believe these updates, guided by your thoughtful review, have substantially improved the manuscript.
>
> Thank you once again for your time and insights.

---

### Official Review · Reviewer_9Jgw · 2024-10-30

**Soundness:** 3
**Presentation:** 3
**Contribution:** 2
**Rating:** 5
**Confidence:** 4

**Summary:**

This paper introduces a novel approach to improve the stability and image quality of drag editing, namely GoodDrag. GoodDrag adopts an AlDD framework that alternates between drag and denoising operations within the diffusion process, effectively improving the fidelity of the result. They also propose an information-preserving motion supervision operation that maintains the original features of the starting point for precise manipulation and artifact reduction. In addition, they contribute to the benchmarking of drag editing by introducing a new dataset, Drag100, and developing dedicated quality assessment metrics, Dragging Accuracy Index and Gemini Score, utilizing Large Multimodal Models.

**Strengths:**

1. The paper proposes an effective framework for achieving high-quality drag-based image editing results.
2. The results demonstrate a significant improvement in image fidelity during the drag editing process.
3. The authors introduce a new dataset and additional metrics for evaluating the point-based image editing task.

**Weaknesses:**

1. The novelty of the ALDD design is limited, as it mainly alters the order of denoising during the drag updating process. A concern is whether separating the process using the hyperparameter 'B' is optimal. Is 'B' fixed, or does it require manual adjustment by the user?
2. There is confusion about whether "ALDD" and "AlDD" refer to the same concept.
3. There is an unnecessary blank area on Line 054 of the paper.
4. The authors claim that previous datasets are limited in terms of diverse drag tasks. However, the proposed Drag100 dataset contains fewer images compared to DragBench (205 images) [1].
5. Providing results for the GoodDrag method on the DragBench dataset[1] would enhance the evaluation.
6. It would be helpful to see how the results are measured using Image Fidelity and Mean Distance metrics in DragDiffusion[1].

[1] Shi, Yujun, et al. "Dragdiffusion: Harnessing diffusion models for interactive point-based image editing." CVPR. 2024.

**Questions:**

1. The paper may lack a discussion on its limitations.

---

> ### Author Response · Authors · 2024-11-27
> **Response to Reviewer 9Jgw**
>
> We thank the reviewer for the valuable comments on this paper and address the raised issues here and in the revised manuscript.
>
> **Q1: The novelty of the AlDD design is limited, as it mainly alters the order of denoising during the drag updating process.**
>
> **A1**: While the operation of AlDD may appear simple at first glance, its development required new insights and deep understanding of the underlying problem, and no prior work has proposed this approach. Notably, we for the first time demonstrate that a subtle adjustment in the denoising order can have a substantial impact on the drag editing process, which is by no means straightforward to identify or realize.
>
> As evidenced in Figure 4, Figure 14, and Table 1, AlDD effectively improves the accuracy and quality of drag editing. Furthermore, GoodDrag, incorporating AlDD, achieves superior performance compared to SOTA methods across multiple datasets, including Drag100 and DragBench, which is demonstrated both qualitatively and quantitatively across various metrics, such as DAI, GScore, and MD, as shown in Figure 7–10, Figure 12–13, and Table 1–10. These results underscore the importance of AlDD, highlighting its innovative contribution to the field and its practical effectiveness in delivering state-of-the-art drag editing performance.
>
> **Q2: How to choose hyperparameter B**
>
> **A2**: As explained in our paper (Figure 3),  $K$ represents the total number of drag operations, and $B$ denotes the number of drag operations performed before each denoising step during AlDD. Consequently, the total number of denoising steps in the AlDD phase is  $K / B$, and the range of $B$ is constrained by  $1 \leq K / B \leq T$ , where $T$ is the starting timestep, representing the total number of denoising steps in the drag editing process.
>
> As introduced in Section 5.1, we set $T = 38$ and $K = 70$, which gives the range $2 \leq B \leq 70$. When $B = 70$, the process reverts to the baseline case without AlDD, where all drag operations are performed at a single timestep ($t = T$).
>
> In general, $K / B$ should not be too large, as drag operations performed at very low timesteps ($t$ close to 0) occur in a latent space that closely resembles the image space, which is unreliable for significant edits. We experimented with various values of $B$ and found that $B = 10$ (resulting in $K / B = 7$ denoising steps in AlDD) consistently yields desirable drag editing results. Therefore, we use a fixed $B = 10$ throughout all our experiments.
>
> **Q3: “ALDD” and “AlDD”**
>
> **A3**: Thank you for pointing out the inconsistency in naming. “ALDD” was a typo error which we have corrected to “AlDD” in the revised manuscript.
>
> **Q4: Blank area on Line 054**
>
> **A4**: Thank you for pointing this out. We have revised the formatting to remove the blank area on Line 054 in the updated manuscript.
>
> **Q5: Diverse drag tasks compared to DragBench**
>
> **A5**: While DragBench contains more images, we argue that diversity cannot be solely measured by the number of images. As introduced in Section 4.1 of the main paper, our insight lies in recognizing the variety of drag editing tasks, which includes relocation, rotation, rescaling, content removal, and content creation, as illustrated in Figure 6. These tasks encompass distinct characteristics that DragBench does not explicitly address. Relocation involves moving an object or a part of an object, while rotation adjusts the orientation of objects; both tasks mimic rigid motion in the physical world without changing the object area or creating new contents. Rescaling corresponds to enlarging or shrinking an object. Content removal involves deletion of specific image components, such as closing a mouth, whereas content creation involves generating new content not present in the original image, such as opening a mouth. By thoughtfully incorporating samples of all different tasks, Drag100 prioritizes task diversity over sheer quantity, offering a more comprehensive and balanced evaluation framework for drag editing algorithms.
>
> **Q6: Results on DragBench**
>
> **A6**: The results on DragBench are provided in Appendix B and C (Tables 3–6). Our GoodDrag demonstrates favorable performance compared to state-of-the-art methods, consistently achieving superior results across different metrics, including DAI, GScore, and MD.

---

> > ### Author Response · Authors · 2024-11-27
> > **Response to Reviewer 9Jgw**
> >
> > **Q7: Evaluation with IF and MD**
> >
> > **A7**: We provide the MD and IF results in Table 5 and 6 of the revised Appendix C, and also include them below for easier reference. Our method achieves better MD values than the baseline methods, demonstrating its effectiveness in drag editing. Note that while MD can well measure drag accuracy, it is considerably slower than the proposed DAI as detailed in Appendix C.
> >
> >    | MD ($\downarrow$)       	| Ours  | DragDiffusion | SDEDrag | DragonDiffusion |
> >    |------------------|-------|---------------|---------|-----------------|
> >    | DragBench    	| **23.40** | 33.50     	| 47.84   | 27.04       	|
> >    | Drag100      	| **23.44** | 37.20     	| 74.33   | 28.40       	|
> >
> >  | IF ($\uparrow$)       	| Ours | DragDiffusion | SDEDrag | DragonDiffusion |
> >    |------------------|------|---------------|---------|-----------------|
> >    | DragBench    	| 0.87 | 0.88      	| **0.91**	| 0.90        	|
> >    | Drag100      	| 0.86 | 0.87      	| **0.89**	| 0.88        	|
> >
> > While the IF score of our method is slightly lower than other approaches, we argue that the IF metric is fundamentally flawed as an evaluation measure for drag editing approaches. IF is defined as 1-LPIPS between the drag-edited image and the input image, meaning it penalizes any changes to the image, even when such changes are necessary to achieve the desired editing. As a result, the metric rewards outputs that are identical or nearly identical to the input image, which contradicts the very purpose of drag editing.
> >
> > This inherent limitation is clearly demonstrated in Figure 11 of the revised Appendix C, where an example with superior visual quality receives the worst IF score, underscoring its failure to reflect meaningful drag edits. In contrast, the proposed GScore metric better correlates with human perception, making it a more reliable and appropriate metric for evaluating drag editing algorithms. Please refer to Appendix C for more details.

---

> ### Author Response · Authors · 2024-11-30
>
> Dear Reviewer 9Jgw,
>
> As the discussion phase is coming to a close, we wanted to check if our responses have fully addressed your concerns. If there are any remaining issues that need clarification, please let us know—we’d be happy to provide additional details.
>
> Thank you!

---

> ### Author Response · Authors · 2024-12-01
> **Follow-Up on Rebuttal Feedback**
>
> Dear Reviewer 9Jgw,
>
> Thank you for your valuable and constructive feedback on our submission. Your comments have been instrumental in improving the quality of our manuscript.
>
> As the discussion deadline approaches, we want to ensure that all your concerns have been fully addressed. Please feel free to reach out if you require further clarifications or additional analyses.
>
> To address your points:
> - We elaborate on the unique contribution of AlDD, demonstrating that a subtle adjustment in the denoising order can have a substantial impact on drag editing — an insight that no prior work has explored or shown.
>
> - We provide additional details on how to select the hyperparameter $B$.
>
> - We highlight the task diversity of Drag100, showcasing how it goes beyond DragBench by encompassing a broader range of editing tasks.
>
> - Results for DragBench with MD and IF are in Appendices B and C, with comparisons for DAI, GScore, MD, and IF detailed in Tables 1–6.
>
> - The paper’s organization has been updated as follows: 1) A typo error “ALDD” was corrected to “AlDD”. 2) Additional space in Lines 54 have been removed as suggested.
>
> - A new section has been added in Appendix N to discuss the limitations of our work.
>
> We hope these updates address your concerns and reflect the improvements made to the paper. If you have had a chance to review our rebuttal and revisions, we would greatly appreciate your feedback or confirmation on whether the updated manuscript meets your expectations.
>
> Additionally, we hope you might consider raising your score based on the revisions, as we believe these updates, guided by your thoughtful review, have substantially improved the manuscript.
>
> Thank you once again for your time and insights.

---

### Official Review · Reviewer_dZPL · 2024-11-02

**Soundness:** 1
**Presentation:** 2
**Contribution:** 1
**Rating:** 3
**Confidence:** 3

**Summary:**

This paper introduces a new approach for enhancing stability and image quality in drag editing tasks. Key contributions include: Alternating Drag and Denoising (AlDD) alternates between drag and denoising steps to prevent artifacts and ensure higher fidelity. Information-Preserving Motion Supervision maintains feature consistency and reduce drifting issues. Experiments are conducted on Drag100 dataset.

**Strengths:**

1. This paper contributes a new dataset and metric for image editting.
2. The idea for solving accumulation distortion is interesting.

**Weaknesses:**

1. This paper does not include comparisons with recent works [1, 2], which limits the context for understanding its contributions relative to the latest advancements.

2. It lacks critical experiments on the DragBench dataset [3], especially with the MD and IF metrics, which are widely used in existing literature for standardized evaluation.

Due to above considertation, it is challenging to assess the performance of this paper.

[1] Drag your noise: Interactive point-based editing via diffusion semantic propagation

[2] EasyDrag: Efficient Point-based Manipulation on Diffusion Models

[3] Dragdiffusion: Harnessing diffusion models for interactive point-based image editing.

**Questions:**

See the strengths and weaknesses

---

> ### Author Response · Authors · 2024-11-27
> **Response to Reviewer dZPL**
>
> We thank the reviewer for the valuable comments on this paper and address the raised issues here and in the revised manuscript.
>
> **Q1: Comparison with more recent works**
>
> **A1**: Thanks for pointing out the related works. We have provided comparisons against more recent methods, including DragNoise and EasyDrag, in the revised Appendix D. These comparisons are presented both qualitatively in Figure 12 and quantitatively in Table 7. The results clearly demonstrate that our proposed method achieves superior performance, which delivers more accurate drag editing, produces images with significantly higher quality, and reduces artifacts compared to the baseline methods. These related works have also been cited in the main paper.
>
> **Q2: Results on DragBench, especially with MD and IF metrics**
>
> **A2**: We present the results on DragBench in Appendices B and C, where the proposed GoodDrag significantly outperforms baseline approaches on both Drag100 and DragBench datasets across the DAI, GScore, and MD metrics.
>
> Detailed comparisons for DAI and GScore on Drag100 and DragBench are provided in Table 1-4 of our paper. The MD and IF results are provided in Table 5 and 6 of the revised Appendix C, and we also include them below for easier reference:
>
>    | MD ($\downarrow$)       	| Ours  | DragDiffusion | SDEDrag | DragonDiffusion |
>    |------------------|-------|---------------|---------|-----------------|
>    | DragBench    	| **23.40** | 33.50     	| 47.84   | 27.04       	|
>    | Drag100      	| **23.44** | 37.20     	| 74.33   | 28.40       	|
>
>  | IF ($\uparrow$)       	| Ours | DragDiffusion | SDEDrag | DragonDiffusion |
>    |------------------|------|---------------|---------|-----------------|
>    | DragBench    	| 0.87 | 0.88      	| **0.91**	| 0.90        	|
>    | Drag100      	| 0.86 | 0.87      	| **0.89**	| 0.88        	|
>
> While the IF score of our method is slightly lower than other approaches, we argue that the IF metric is fundamentally flawed as an evaluation measure for drag editing approaches. IF is defined as 1-LPIPS between the drag-edited image and the input image, meaning it penalizes any changes to the image, even when such changes are necessary to achieve the desired editing. As a result, the metric rewards outputs that are identical or nearly identical to the input image, which contradicts the very purpose of drag editing.
>
> This inherent limitation is clearly demonstrated in Figure 11 of the revised Appendix C, where an example with superior visual quality receives the worst IF score, underscoring its failure to reflect meaningful drag edits. In contrast, the proposed GScore metric better correlates with human perception, making it a more reliable and appropriate metric for evaluating drag editing algorithms.

---

> ### Author Response · Authors · 2024-11-30
>
> Dear Reviewer dZPL,
>
> As the discussion phase is coming to a close, we wanted to check if our responses have fully addressed your concerns. If there are any remaining issues that need clarification, please let us know—we’d be happy to provide additional details.
>
> Thank you!

---

> ### Author Response · Authors · 2024-12-01
> **Follow-Up on Rebuttal Feedback**
>
> Dear Reviewer dZPL,
>
> Thank you for your valuable and constructive feedback on our submission. Your comments have been instrumental in improving the quality of our manuscript.
>
> As the discussion deadline approaches, we want to ensure that all your concerns have been fully addressed. Please feel free to reach out if you require further clarifications or additional analyses.
>
> To address your points:
> - Comparisons with recent methods, including DragNoise and EasyDrag, are provided in Appendix D, with qualitative results in Figure 12 and quantitative results in Table 7.
>
> - Results for DragBench with MD and IF are in Appendices B and C, with comparisons for DAI, GScore, MD, and IF detailed in Tables 1–6.
>
> We hope these updates address your concerns and reflect the improvements made to the paper. If you have had a chance to review our rebuttal and revisions, we would greatly appreciate your feedback or confirmation on whether the updated manuscript meets your expectations.
>
> Additionally, we hope you might consider raising your score based on the revisions, as we believe these updates, guided by your thoughtful review, have substantially improved the manuscript.
>
> Thank you once again for your time and insights.

---

### Official Review · Reviewer_jGsQ · 2024-11-02

**Soundness:** 3
**Presentation:** 3
**Contribution:** 4
**Rating:** 8
**Confidence:** 4

**Summary:**

The paper introduces a drag-editing framework using diffusion. The paper comprehensively evaluates and fixes issues in prior work (For ex- introduces information preservation through the feature alignment loss to alleviate drag points' drifting issue). Overall, the framework proposes a simple re-ordering of existing control and denoising steps leading to no additional computational overhead while significantly improving editing fidelity.

The paper also contributes a dataset (Drag100) geared towards editing evaluation rather than training, in addition to two evaluation metrics for measuring editing fidelity. One of the metric exploits

**Strengths:**

1. Paper comprehensively evaluates prior work and builds on the weaknesses through a simple alternating dragging & denoising step-framework.
2. The paper introduces a carefully thought out (natural + synthetic) hybrid evaluation dataset comprising diverse editing scenarios; in addition to two evaluation metrics that might be useful for the drag-editing community. Potential for higher impact.
3. Results look promising and the information preservation feature alignment loss is well grounded.
4. Supplementary contains substantial ablations and empirical visual justifications for proposed components of the framework.

**Weaknesses:**

1. It makes sense to compare against diffusion based editing techniques since the paper builds on the weaknesses of prior literature. However, it would be nice to still have DragGAN in the main qualitative figure (7) and the quantitative evaluations for the sake of completeness.
2. Training hardware and memory is not specified in the main text. Consider moving it from the supplementary to the main text?

**Questions:**

Suggestions:
1. I strongly recommend moving the user study carried out by the authors (in the supplementary currently) to the main text since that is the common evaluation metric across different papers, not DAI or GScore.
2. L355 - 357 is not needed. It should be apparent to the readers in this (editing/generative) space.
3. Consider making figures bigger. Especially Figures 5, 6 and 7.
4. Add an ethics/societal impact section.

If these minor weaknesses and suggestions are acknowledged, I would consider giving it an even higher rating for highlighting.

---

> ### Author Response · Authors · 2024-11-27
> **Response to Reviewer jGsQ**
>
> We thank the reviewer for the valuable comments on this paper and address the raised issues here and in the revised manuscript.
>
> **Q1: Comparison with DragGAN**
>
> **A1**: We focus on comparisons with diffusion-based methods in Figure 7 and provide comparisons with DragGAN in Figure 8. In response to the reviewer’s suggestion, we have included DragGAN’s results for the examples in Figure 7 in Figure 13 of the revised Appendix E. Additionally, quantitative results for DragGAN are presented in Table 8 of the revised Appendix E.
>
> It is important to note that DragGAN has significant limitations in terms of efficiency, as it requires fine-tuning the GAN generator for each new input image, resulting in a much slower process compared to our method. Due to the tight timeline of the rebuttal phase, we were only able to evaluate DragGAN on the six images in Figure 7. Nevertheless, our results demonstrate that GoodDrag achieves superior performance both qualitatively and quantitatively. We plan to conduct a more comprehensive evaluation, including additional examples and detailed performance comparisons, in the final version of the paper.
>
>
> **Q2: Moving training hardware and memory to main text**
>
> **A2**: Thanks for the suggestion. We have moved the details of the training hardware and memory usage to Section 5.1 of the revised main paper.
>
> **Q3: Moving user study to main text**
>
> **A3**: Thanks for the suggestion. We have moved the user study to Section 5.2 and Figure 9 of the revised main paper.
>
> **Q4: L355-357 is not needed**
>
> **A4**: Thank you for the suggestion. We have removed these lines as suggested.
>
> **Q5: Making figures bigger**
>
> **A5**: We have enlarged Figure 5, 6, and 7 for better readability.
>
> **Q6: Adding an ethics/societal impact section**
>
> **A6**: Following the suggestion, we have added a new section in Appendix O discussing the potential societal impact of our work.

---

> ### Author Response · Authors · 2024-11-30
>
> Dear Reviewer jGsQ,
>
> As the discussion phase is coming to a close, we wanted to check if our responses have fully addressed your concerns. If there are any remaining issues that need clarification, please let us know—we’d be happy to provide additional details.
>
> Thank you!

---

> > ### Author Response · Authors · 2024-12-01
> > **Follow-Up on Rebuttal Feedback**
> >
> > Dear Reviewer jGsQ,
> >
> > Thank you for your valuable and constructive feedback on our submission. Your comments have been instrumental in improving the quality of our manuscript.
> >
> > As the discussion deadline approaches, we want to ensure that all your concerns have been fully addressed. Please feel free to reach out if you require further clarifications or additional analyses.
> >
> > To address your points:
> > - DragGAN’s results have been added to Figure 7, along with quantitative results presented in Table 8 of the revised Appendix E.
> >
> > - The paper’s organization has been updated as follows: 1) Hardware details have been moved to Section 5.1 of the revised main paper. 2) The user study is now included in Section 5.2 and illustrated in Figure 9 of the main paper. 3) Lines 355–357 have been removed as suggested. 4) Figures 5, 6, and 7 have been resized for improved readability.
> >
> > - A new section has been added in Appendix O to discuss the potential societal impact of our work.
> >
> > We hope these updates address your concerns and reflect the improvements made to the paper. If you have had a chance to review our rebuttal and revisions, we would greatly appreciate your feedback or confirmation on whether the updated manuscript meets your expectations.
> >
> > Additionally, we hope you might consider raising your score based on the revisions, as we believe these updates, guided by your thoughtful review, have substantially improved the manuscript.
> >
> > Thank you once again for your time and insights.

---

### Meta-Review · Area_Chair_s93t · 2024-12-17

**Metareview:**

This paper introduces GoodDrag, a novel approach to improve the stability and image quality of drag editing. It alternates between drag and denoising operations within the diffusion process, improving the fidelity of the result. It proposes an information-preserving motion supervision operation that maintains the original features of the starting point for precise manipulation and artifact reduction. It also introduces a new dataset (Drag100) and develops dedicated quality assessment metrics utilizing Large Multimodal Models. Extensive experiments demonstrate that GoodDrag compares favorably against state-of-the-art approaches both qualitatively and quantitatively.

Strengths:
- The paper comprehensively evaluates prior work and builds on the weaknesses through a simple alternating dragging & denoising step-framework.
- The paper introduces a carefully thought out (natural + synthetic) hybrid evaluation dataset comprising diverse editing scenarios; in addition to two evaluation metrics that might be useful for the drag-editing community.
- Results look promising and the information preservation feature alignment loss is well-grounded.
- Supplementary contains substantial ablations and empirical visual justifications for proposed components of the framework.

Weaknesses:
- Some presentation issues
- Limited comparison to recent work and other datasets (DragBench) in the original submission.
- Method's limitation was not originally discussed in the submission.
- Limited novelty; unclear contributions (pointer out by one reviewer)

After the rebuttal most of the concerns have been addressed. As a summary:
- The paper addresses an important problem in drag editing with diffusion models: the accumulation of perturbations and resulting distortions.
- The proposed GoodDrag approach is novel and technically sound.
- The experimental results are promising and demonstrate the effectiveness of the proposed approach.
- The paper is well-written and easy to follow.

**Additional Comments On Reviewer Discussion:**

The authors rebuttal helped clarify many of the reviewers concern. In particular:

- Reviewer jGsQ recommended moving the user study to the main text and adding an ethics/societal impact section.  The authors responded by moving the user study to the main text and adding an ethics/societal impact section.
- Reviewer dZPL pointed out that the paper did not include comparisons with recent works or critical experiments on the DragBench dataset.  The authors responded by adding comparisons with recent works and providing results on the DragBench dataset.
- Reviewer 9Jgw questioned the novelty of the AIDD design and asked for clarification on the hyperparameter 'B'.  The authors responded by elaborating on the novelty of AIDD and providing additional details on how to select the hyperparameter 'B'.
- Reviewer k1h1 asked for additional experimental results, including the impact of using different base models and a discussion of limitations.  The authors responded by providing additional experimental results and discussing the limitations of their work.
- Reviewer pphp questioned the novelty of AIDD and asked for a running time analysis.  The authors responded by clarifying the novelty of AIDD and providing a running time analysis.

---

### Decision · Program_Chairs · 2025-01-22

Accept (Poster)